# On the Mechanism of Heterogeneous Water Oxidation Catalysis: A Theoretical Perspective

**Shanti Gopal Patra** [1,*] and **Dan Meyerstein** [2,3,*]

1    Department of Chemistry, Indian Institute of Technology, Kharagpur 721302, India
2    Department of Chemical Sciences, The Radical Research Center, Ariel University, Ariel 40700, Israel
3    Department of Chemistry, Ben-Gurion University, Beer-Sheva 8410501, Israel
*    Correspondence: patrashantigopal@gmail.com (S.G.P.); danm@ariel.ac.il (D.M.)

**Abstract:** Earth abundant transition metal oxides are low-cost promising catalysts for the oxygen evolution reaction (OER). Many transition metal oxides have shown higher OER activity than the noble metal oxides ($RuO_2$ and $IrO_2$). Many experimental and theoretical studies have been performed to understand the mechanism of OER. In this review article we have considered four earth abundant transition metal oxides, namely, titanium oxide ($TiO_2$), manganese oxide/hydroxide ($MnO_x$/MnOOH), cobalt oxide/hydroxide ($CoO_x$/CoOOH), and nickel oxide/hydroxide ($NiO_x$/NiOOH). The OER mechanism on three polymorphs of $TiO_2$: $TiO_2$ rutile (110), anatase (101), and brookite (210) are summarized. It is discussed that the surface peroxo O* intermediates formation required a smaller activation barrier compared to the dangling O* intermediates. Manganese-based oxide material $CaMn_4O_5$ is the active site of photosystem II where OER takes place in nature. The commonly known polymorphs of $MnO_2$; α-(tetragonal), β-(tetragonal), and δ-(triclinic) are discussed for their OER activity. The electrochemical activity of electrochemically synthesized induced layer δ-$MnO_2$ (EI-δ-$MnO_2$) materials is discussed in comparison to precious metal oxides (Ir/$RuO_x$). Hydrothermally synthesized α-$MnO_2$ shows higher activity than δ-$MnO_2$. The OER activity of different bulk oxide phases: (a) $Mn_3O_4$(001), (b) $Mn_2O_3$(110), and (c) $MnO_2$(110) are comparatively discussed. Different crystalline phases of CoOOH and NiOOH are discussed considering different surfaces for the catalytic activity. In some cases, the effects of doping with other metals (e.g., doping of Fe to NiOOH) are discussed.

**Keywords:** heterogeneous OER; DFT+U; titanium oxide; manganese oxide; cobalt oxide; nickel oxide

## 1. Introduction

The oxidation process involves the following reactions: [1–3]
In acidic medium:

$$2H_2O\ (l) \rightarrow O_2 + 4H^+ + 4e^- \qquad E° = 1.23\ \text{V vs. NHE} \qquad (1)$$

$$4H^+ + 4e^- \rightarrow 2H_2\ (g) \qquad E° = 0.00\ \text{V vs. NHE} \qquad (2)$$

In basic medium:

$$4OH^-\ (l) \rightarrow O_2\ (g) + 2H_2O\ (l) + 4e^- \qquad E° = 0.40\ \text{V vs. NHE} \qquad (3)$$

$$2H_2O\ (l) + 4e^- \rightarrow 2H_2\ (g) + 4OH^-\ (l) \qquad E° = -0.83\ \text{V vs. NHE} \qquad (4)$$

The oxidation step involves four electrons and four protons and hence requires high potential (1.23 V). However, in practice the formal potential for the water oxidation is even higher than the standard one and this difference is termed as overpotential. The challenges to the synthetic and electrochemist is to develop catalysts that would require smaller overpotentials. Nature smartly utilizes this process in photosystem II to withdraw protons and electrons and releases oxygen [4]. Another implication of the water splitting

reaction is the generation of hydrogen, which can be utilized as fuel and hence as an alternative source to fossil fuel. As the process of oxygen evolution reaction (OER) involves protons, it is essentially a proton coupled electron transfer reaction (PCET) and hence pH dependent [5–10]. Thus, following the Nernst equations, the water oxidation potential can be substantially reduced in alkaline medium.

The water oxidation process is catalyzed by molecular catalysts as well as by heterogeneous catalysts. In the past two decades a plethora of transition metal-based molecular catalysts of Mn [11–14], Fe [15–17], Co [18], Ni [19–21], Cu [22–31], Ru [32–34], etc., have been developed [1,35]. The crucial step in the water oxidation mechanism is the peroxo linkage formation, which involves either water nucleophilic attack (Equation (5)) or radical recombination (Equations (6)–(8)) [36].

$$L_1M^n\!=\!O + H_2O \rightarrow L_1M^{n-2}-OOH + H^+ \tag{5}$$

$$2L_1M^n\!=\!O \rightarrow L_1M^{n-1}-OO-M^{n-1}L_1 \tag{6}$$

$$L_1M^n-(OH)_2 \rightarrow L_1M^{n-2}-(O_2) + 2H^+ \tag{7}$$

$$2L_1M^n-OH \rightarrow L_1M^{n-1}-OO-M^{n-1}L_1 + 2H^+ \tag{8}$$

Materials that have shown promising heterogeneous OER activity are platinum surfaces, transition metal oxides, first-row transition metal spinels, perovskites, carbonates, and metal organic framework (MOF) [37]. The efficiency of the heterogeneous catalyst depends on low coordinate sites, which are defects on the surfaces that are tuned by applying various synthetic procedures. The platinum metal surface has been the most active and widely studied catalyst for the OER, which shows an exchange current density of $10^{-9}$ A/cm$^2$ [38]. However, before the catalytic oxidation can start, the Pt surface must be converted to oxide. This results in a time lag in the oxygen evolution process. The kinetics are understood though the Tafel slope, which suggests that for the low current density the mechanism is independent of film thickness. However, for the high current density it depends on the thickness. Iridium oxide (IrO$_2$) is used in polymer electrolyte, which has industry standard performance. Ruthenium oxide (RuO$_2$) is the most promising OER catalyst for an acidic environment [37]. However, Ru, Ir, and Pt are precious metals and hence are not cost effective. Thus, scientists are involved in developing low cost, highly abundant first row transition metal-based catalysts.

The *Pourbaix Atlas* diagram provides the electrochemical stabilities of the metals at various pHs against corrosion [1]. It was shown that the earth abundant metal oxides are stable at high pH. To separate the cathode and anode compartment, an anion exchange membrane (AEM) is used instead of Nafion [39]. In that way the device is stable in harsh pH conditions and thus the efficiency, stability, and scalability are obtained. Commercial electrolyser uses Pt cathode (for HER) and IRO$_2$ anode (for OER), which are separated by a membrane [40]. It was shown that IrO$_2$ can serve as both a photoanode and water oxidation catalyst under UV and visible light [41].

In the past few years many theoretical chemists have been involved in understanding the mechanism of heterogeneous water oxidation through computational chemistry. Here we summarize the theoretical understanding of the mechanism of OER with regards to earth abundant first row transition metals.

## 2. Computational Methodology

In chemistry, the ground state total energy of a system with a given coordinate is best described by quantum mechanics methods, namely, density functional theory (DFT) [42,43]. In DFT, the total energy is expressed as a function of electronic density $\rho(r)$:

$$E[\rho(r)] = T[\rho(r)] + J[\rho(r)] + V_{\text{ext}[\rho(r)]} + E_{XC}[\rho(r)], \tag{9}$$

where $V_{ext}$ is the potential energy of the nucleus–electron interaction, J is the electron–electron Coulomb repulsion energy terms, $T$ is the kinetic energy of ideal noninteracting electrons, and $E_{xc}$, is the exchange–correlation energy that describes all nonclassical corrections to the electron—electron interactions. In heterogeneous catalysis, the generalized-gradient-approximation (GGA) is employed using the Perdew–Burke–Ernzerhof (PBE) functional [44].

A standard DFT functional, e.g., PBE, cannot describe the strongly correlated 3D metal oxides. Due to the delocalization error, the standard DFT functional underestimates the band gap. This error has been corrected using HSE06 [45], PBE0 [46], and PBE+U [46] functionals. The computational cost of PBE+U is comparable to PBE functional; however, it is high with hybrid functionals PBE0 and HSE06. In the PBE+U scheme to the Hamiltonian's electron–electron interaction potential, the Hubbard term, $U_{eff} = U - J$, is introduced by Dudarev et al. [47] The thermochemistry of the reactions were obtained following the method of Nørskov et al. [48] In this model, the barrier is obtained from the difference in free energies between the intermediates, and the kinetic energy barrier was not calculated. Valdés et al. [49] studied the interaction of water with $TiO_2$ through the analysis of relative stabilities of different surface termination. The method is summarized below:

(1) The reference potential to be that of NHE is considered when $H^+ + e^-$ are involved. At standard conditions ($U = 0$, pH = 0, $p = 1$ bar, $T = 298$ K), the reaction $*AH \rightarrow A + H^+ + e^-$ is equivalent to $*AH \rightarrow A + 1/2\ H_2$ and $\Delta G_0 = \Delta G$ because the free energy of $H^+(aq) + e^-$ can be taken equal to that of $1/2\ H_2$;

(2) For the equation, $\Delta G_0 = \Delta E + \Delta ZPE - T\Delta S$, the $\Delta ZPE$ (zero-point energy) and $\Delta S$ are obtained from the vibrational frequency calculations and the standard free energy table. For the atoms and molecules adsorbed to the coordinatively unsaturated sites (CUSs), the $S = 0$ is considered. The temperature dependence is ignored in the calculations;

(3) Using the $\Delta G_U = -eU$, where $U$ is the electrode potential relative to the standard hydrogen electrode, the potential bias is included in the calculation.

(4) pH correction is done using $\Delta G_{pH}(pH) = -kT \ln 10 \cdot pH$ equation.

(5) The energy values of $H_2O$ and $H_2$ in the gas phase are used as the reference states. At an equilibrium pressure of 0.035 bar, the entropy of gas phase water occurs at room temperature. The step involving $O_2$, $\Delta G_{\{2H2O \rightarrow O2+2H2\}} = 4.92$ eV $= E_{O2} + 2E_{H2} - 2E_{H2O} + (\Delta ZPE - T\Delta S)_{\{2H2O \rightarrow O2+2H2\}}$ is considered;

(6) It was considered that the interaction energy is equal to the energy of a hydrogen bond for the interaction of O*, OH*, OOH*, or an empty CUS site. Further, it was found to be negligible in rutile-type oxides [50].

Plane-wave implementation within DFT was used at the GGA RPBE level [51,52]. For the most stable rutile-type (110) surface, a periodically repeating four-layer slab was chosen. A vacuum of 20 Å was used to separate the slab. For slab calculations, the $2 \times 2 \times 1$ Monkhorst−Pack type of k-point sampling and $2 \times 1$ surface unit cell were utilized. The bottom layer of the slab was fixed at the bulk lattice constant, but the two top layers and the species adsorbed on it were relaxed. To deal with the core, the ultrasoft pseudopotential was used [53]. A plane wave basis set with a cutoff energy of 340 eV was used to represent the electronic wave functions. On a grid, a plane wave cut off of 500 eV was used to treat the electron density. To ensure convergence, the Pulay mixing and Fermi smearing of 0.1 eV were used. All calculations were performed using the atomic simulation environment (ASE) package [54].

Wang et al. [55] have investigated the photocatalytic OER mechanism using extensive first-principles molecular dynamics (MD) simulations at the water/rutile-$TiO_2$(110) interface. All the spin-polarized calculations were performed using VASP code [56,57] with PBE functional. The core-valence electron interaction with electrons from Ti 3$p$, 3$d$, 4$s$; O 2$s$, 2$p$; and H 1$s$ shells explicitly included using the project-augmented wave (PAW) method. With an energy cutoff of 450 eV, the valence electronic state was expanded. Using Gaussian smearing with SIGMA = 0.05 eV, the one-electron states were calculated. Using the

BFGS minimization scheme, the ionic degrees of freedom were relaxed until the Hellman–Feynman forces on each ion were less than 0.05 eV Å$^{-1}$. A constrained optimization scheme was employed to find the transition states. A force threshold of 0.05 eV Å$^{-1}$ was used at the TS. The dipole correction was included in the calculations.

Using a four-Ti-layer $p(1 \times 4)$ periodical rutile (110) slab with a ~15 Å vacuum between slabs and a corresponding $1 \times 2 \times 1$ K-points mesh, the MD calculations were performed. At the liquid/solid interface, a lattice-matching bulk ice (containing 26 $H_2O$) was introduced. The 300 K (experimental temperature) with a 0.5 fs movement for each step in the canonical (NVT) ensemble employing Nosé–Hoover thermostats were employed. It was found that the water structures in our systems often change significantly in the first ~2 ps, and reach quasi-equilibrium around 3 ps. To confirm that the systems were in equilibrium, MD simulation was run for a period of ~6 ps.

Malik et al. [58] considered stoichiometric and symmetric slabs for all the three phases, anatase (101), rutile (110), and brookite (210). Four Ti layers with a vacuum space of 22 Å were considered. The optimized lattice parameters were anatase (101): $a = 10.89$ Å, $b = 15.10$ Å, $c = 27.86$ Å, rutile (110): $\alpha = \beta = 90.00°$, $\gamma = 110.30°$ for $a = 5.91$ Å, $b = 12.99$ Å, $c = 22.28$ Å, and $\alpha = \beta = \gamma = 90.00°$ for and brookite (210): $a = 10.27$ Å, $b = 14.24$ Å, $c = 28.99$ Å and $\alpha = \beta = \gamma = 90.00°$. The top three layers were allowed to relax during optimization. For a sampling of the Brillouin zone based on the Monkhorst–Pack scheme, Γ-centered k-point meshes were used. In all cases, the k-point meshes were $3 \times 2 \times 1$.

Using the linear response theory, the effective $U$–$J$ terms ($U_{eff}$) term was set as 4.0 eV for Mn [59]. In Mn the calculations were performed using PBE+U and HSE06 methods [60]. In the OER mechanism the long-range electrostatic interaction was considered for the solvation energy by a periodic continuum solvation model with a smooth dielectric function [61–63]. The $4 \times 4 \times 4$ in the Monkhorst–Pack scheme was used for the k-point mesh. The geometry convergence criterion was set as 0.01 eV/Å for the maximal component of force and 0.01 GPa for stress. In Mn, using the PAW method, the ionic cores were considered with valence electrons: Mn $3d$ and $4s$ and O $2s$ and $2p$ [64]. Wave functions were described in a plane-wave basis truncated at 650 eV. Simulations were carried out with cells containing ~2.0 nm thick oxide slabs and 2.5 nm thick vacuum.

The OER reaction pathway by CoCat [65] was investigated using an explicit water solution employing ab initio molecular dynamics (AIMD) simulations [66]. The DFT+U calculations were performed using the Quantum ESPRESSO package [67] in a restricted open-shell Kohn–Sham approach. The Co-O clusters were simulated to understand the interface between water and amorphous CoCat. Water molecules surrounded the clusters within periodic boundary conditions. Clusters were built based on the experimental observations: (i) Co atoms are surrounded by six O atoms as obtained from the X-ray absorption measurements. That means the occurrence of terminal oxygen atoms at the CoCat/water interface. (ii) O atoms placed as $\mu_2$-O bridges between Co(III) ions are likely to be protonated, as suggested by the X-ray absorption spectroscopy (XAS)data. (iii) CoCat is saturated by H atoms. (iv) IR measurements confirm the presence of Co(III)–$OH_2$ and Co(III)–OH species. A U correction of 5.9 eV was applied to the Co atoms obtained from the self-consisted linear response approach.

Bajdich et al. [68] outlined a scheme for the calculations. The elementary steps involving adsorbed OH and O species on the surface (*) follow the scheme:

$$OH^- + * \rightarrow OH^* + e^- \tag{10}$$

$$OH^* + OH^- \rightarrow O^* + H_2O + e^- \tag{11}$$

where $O_2$ is formed via one of two pathways. One is the direct recombination of two O* intermediates as

$$O^* + O^* \rightarrow O_2 \tag{12}$$

The second is the two-step reaction of O* with OH⁻ via OOH* as an intermediate:

$$O^* + OH^- \rightarrow OOH^* + e^- \tag{13}$$

$$OOH^* + OH^- \rightarrow O_2 + H_2O + e^- \tag{14}$$

In acidic medium, Reactions (9), (10), (12), and (13) are modified as:

$$2H_2O + ^* \rightarrow OH^* + H_2O + e^- + H^+ \tag{15}$$

$$OH^* + H_2O \rightarrow O^* + H_2O + e^- + H^+ \tag{16}$$

$$O^* + H_2O \rightarrow OOH^* + e^- + H^+ \tag{17}$$

$$OOH^* \rightarrow O_2 + e^- + H^+ \tag{18}$$

Gibbs free energy change for the Reactions (15)–(18) are expressed as:

$$\Delta G_1 = \Delta G_{OH} - eU + \Delta G_{H+}(pH) \tag{19}$$

$$\Delta G_2 = \Delta G_O - \Delta G_{OH} - eU + \Delta G_{H+}(pH) \tag{20}$$

$$\Delta G_3 = \Delta G_{OOH} - \Delta G_O - eU + \Delta G_{H+}(pH) \tag{21}$$

$$\Delta G_4 = 4.92\ [eV] - \Delta G_{OOH} - eU + \Delta G_{H+}(pH) \tag{22}$$

At standard conditions ($T = 298.15$ K, $P = 1$ bar, pH = 0) $U$ is the potential measured against NHE. The pH dependence of free energy is expressed as $\Delta G_{H+}(pH) = -k_B T \ln(10) \times pH$. In Equations (19)–(22), the Gibbs free energies are dependent of adsorption energies of OH*, O*, and OOH*.

The theoretical overpotential is readily obtained using the following equation:

$$\eta = \max[\Delta G_1, \Delta G_2, \Delta G_3, \Delta G_4]/e - 1.23\ [V] \tag{23}$$

All the DFT calculations by the ASE simulation package and the methodology remain the same as discussed earlier.

In the case of Ni in NiOOH, the $U_{eff}$ lies in the range 4.5–6.5 eV. A predefined amount of the exact Hartree–Fock (HF) exchange is mixed to PBE functional to obtain the PBE0 and HSE06, for example:

$$E_{XC}^{PBE0} = aE_x^{HF} + (1-a)E_X^{PBE} + E_C^{PBE} \tag{24}$$

where the PBE exchange is $E_x^{PBE}$, the exact exchange is $E_x^{HF}$, and the PBE correlation is $E_c^{PBE}$. To screen of the long-range part of the exact exchange is a screened Coulomb potential is added to the long-range part of the exact exchange in HSE06. The mixing coefficient $\alpha = 1/4$ for both PBE0 and HSE06 [46]. The parameters were directly taken from the iron oxide [69].

## 3. Results and Discussion

An enormous effort has been put forward to develop a heterogeneous catalyst with earth abundant, low overpotential, high turnover frequency (TOF), and durability. Many transition metal-based oxides and their combinations have already outperformed the activity of noble metal oxides ($RuO_2$ and $IrO_2$). However, it is still a big challenge to the experimental and theoretical chemist to understand the four-electron transfer reaction on a solid surface. The mechanism of heterogeneous water oxidation is very sensitive to the surface and pH of the medium. In acidic media, water molecules bind to the solid surface with simultaneous removal of one proton and one electron (Figure 1) [70]. In the next step the surface bound OH groups further lose one proton and one electron to form an oxygen atom bound to the surface. Next, two proximal O atoms combine to form dioxygen. Thus, in the whole process starting with two water molecules, the end products are one oxygen molecule, four protons, and four electrons.

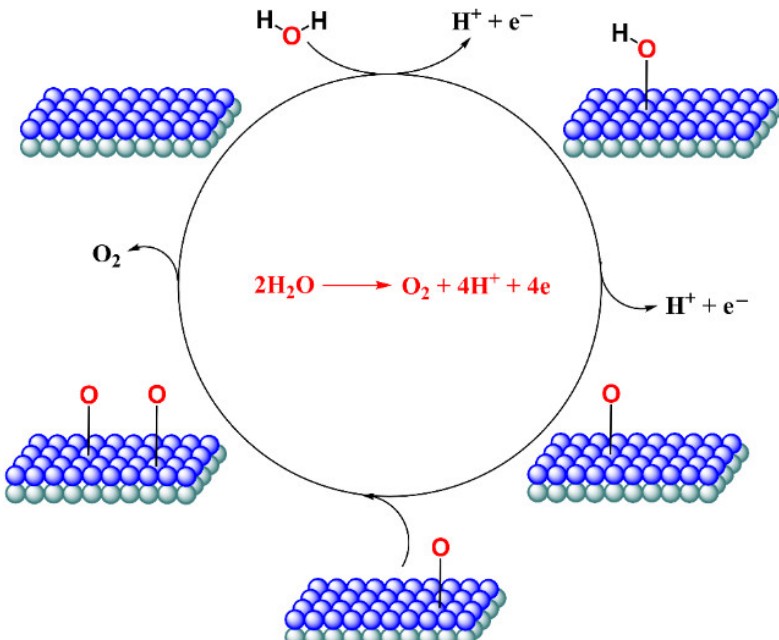

**Figure 1.** Mechanism of heterogeneous water oxidation in acidic condition. Motivated from ref. [70] and Wikipedia (https://en.wikipedia.org/wiki/Heterogeneous_water_oxidation#cite_note-Birss1-5 (accessed on 4 September 2022)).

On the other hand, in alkaline media, instead of water, the hydroxyl ions first get bound to the solid surface (Figure 2) [71]. Followed by the removal of one electron, the OH$^-$ group is converted to an OH group. Then, from the medium, the OH$^-$ ions abstract protons from the metal bound OH group to form water and metal bound O$^-$. In the next step O$^-$ is converted to metal bound O atom. Two closely spaced O atoms react with each other to form oxygen.

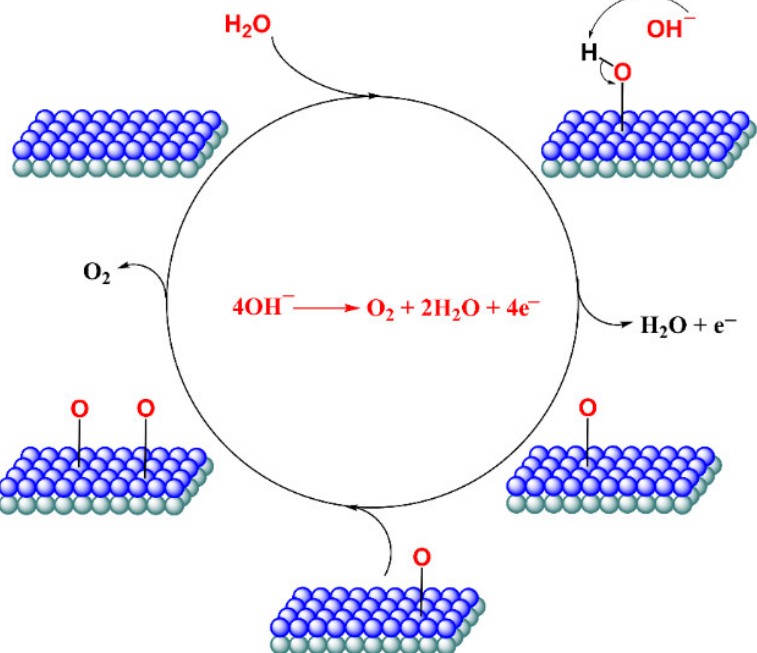

**Figure 2.** Mechanism of heterogeneous water oxidation in basic condition. Motivated from ref. [71] and Wikipedia (https://en.wikipedia.org/wiki/Heterogeneous_water_oxidation#cite_note-Birss1-5 (accessed on 4 September 2022)).

The high overpotential at the anode is a major hurdle of the OER mechanism. The OER mechanism on metal oxides varies with different phases and surfaces of the same material. The OER mechanism on oxide can proceed through a variety of processes (see Section 3.4). The catalytic effect on (photo)anode may arise from the relative stabilities of the reaction intermediates. Studies by various groups have shown that an ideal picture cannot be modeled due to the correlated binding energies of the intermediate [72–75]. The restriction on the possibility of optimizing the binding energies arises due to the scaling relationship. It has been reported that there is a constant difference in the binding energies of hydroxyl and peroxyl at metal or oxide surfaces [2,76]. However, in reality, this value deviates from the ideal one, and that leads to an overpotential of 0.4 V, considering no reaction activation barrier. Mostly spin restricted calculations were performed to deal with the intermediates and scaling relationship. Most of the oxide surfaces are paramagnetic, and the spin is quenched when an adsorbate is placed on the surface. For metals, spin quenching was often found to be incomplete [77,78]. Mom et al. [75] studied the effect of incomplete spin quenching for the estimated catalytic performance of metal oxides.

Rao et al. [79] studied the change in surface structure on varying the potential in an acidic solution. Due to successive deprotonation of water on the coordinatively unsaturated Ru sites (CUS) and hydrogen adsorbed, the redox peaks were observed at 0.7, 1.1, and 1.4 V vs. NHE. At water oxidation, potential peroxide species are formed on the Ru CUS sites, which the neighboring OH group stabilizes. A new pathway was suggested where O-O stabilizes deprotonation of the OH group species at the rate-limiting step. Similar studies were performed on (100), (110), and (101) surfaces of $RuO_2$ [80]. The binding energy decreases from (110) to (100) to (101), and hence the reactivity increases in the same order. In a different rate-limiting step, $-O + H_2O - (H^+ + e^-) \rightarrow -OO-H$, the binding energy is further decreased in the $Ru_{CUS}$ site of (101).

Single-atom catalysts (SACs) have gained much attention for their role in catalytic hydrogen evolution reactions (HER). The activity of SACs was rationalized by the Nørskov model [81] where the free energy of H atom adsorption on the extended metal surface is used to change the exchange current. The SACs are also considered analogous to coordination compounds and not as a metal surfaces. It shows that stable dihydride or dihydrogen complexes (HMH) compounds are formed similar to coordination compounds [82]. It was shown that the formation of MHM in addition to MH may change the kinetics. On SACs, a three-dimensional volcano plot for the HER was obtained by extending the model to two intermediates (MH and HMH). In SACs, the role of the formation of superoxo and peroxo complexes was studied by Capriano et al. [83]. Eleven transition metal atoms (Sc, V, Ti, Cr, Mn, Fe, Co, Ni, Cu, Pd, Pt) were included in the nitrogen-doped graphene in the study. The inclusion of the superoxo/peroxo complexes in the reaction profile is dependent on transition metals. The oxygen reduction reaction is generally considered to be $O_2 \rightarrow OOH^* \rightarrow O^* \rightarrow OH^* \rightarrow H_2O$ (O* mechanism). An unconventional mechanism $O_2 \rightarrow OOH^* \rightarrow 2OH^* \rightarrow OH^* \rightarrow H_2O$ was discussed by Zhong et al. [84] using single-atom catalysts (Me/N/C, Me = Fe, Co, etc.). The structure dependence of oxygen reduction reaction (ORR) and its heuristic nature were discussed in detail.

### 3.1. Titanium Oxide-Based Catalyst

The ground-breaking work on photo-electrocatalysis of water using $TiO_2$ by Fujishima and Honda [85] gained much attention and led to the development of semiconducting materials for photo-electrocatalysis. It serves as a standard material to understand photo oxidation despite having a large band gap that is higher than other commonly used semiconducting materials such as $WO_3$, $Fe_2O_3$, or $TiO_2$ [86,87]. As the photoenergy conversion by $TiO_2$ is out of the visible region, it has less than 1% solar photo energy conversion factor [88]. Thus, scientists are constantly engaged in developing visible energy-based photo-electrocatalyst to increase the energy conversion factor. Low cost, low toxicity, and durability under illuminations are the merits of $TiO_2$. Via doping of other metals and using $TiO_2$ nanotube arrays the activity can be extended from UV to the visible region. The

trend in activity with respect to other rutile type oxides is $RuO_2 < IrO_2 < TiO_2$ with an overpotential of 1.19 V for $TiO_2$.

Valdés et al. [49] studied the interaction of water with $TiO_2$ through the analysis of relative stabilities of different surface termination. The key step is the reaction of water with the surface and formation of hydroxyl groups (OH*) bound to the surface. Two types of surfaces are considered: (i) $S^1$ ($TiO_2 + 2O_b$), with all bridge sites occupied by oxygen; and (ii) $S^2$ ($TiO_2 + 2O_b + 2O*$), totally O covered surface (O* represents surface adsorbed oxygen to a CUS site) among five different possible surfaces (Figure 3). In $S^2$ the coordinatively unsaturated sites (CUS) are also occupied by oxygen.

| $S^1$ | $S^2$ | $S^3$ | $S^4$ | $S^5$ |
|---|---|---|---|---|
| $\Delta G = 0$ eV | $\Delta G = 7.40$ eV $-4eU$ $-4$ kT ln10 pH | $\Delta G = 4.58$ eV $-2eU$ $-2$ kT ln10 pH | $\Delta G = 0.87$ eV | $\Delta G = 0.53$ eV |

**Figure 3.** The structures of various surfaces studied. $S^1$, $S^2$, $S^3$, $S^4$, and $S^5$ represent the surface with oxygen occupying all bridge sites occupied by oxygen, the surface with totally covered O*, the surface with bridge sites occupied oxygen and CUS sites occupied by HO*, the surface totally covered by HO*, and the surface with alternating deprotonated non-deprotonated adsorbed water. The blue, red, and white spheres represent metal ions, oxygen, and hydrogen, respectively. (Reprinted from ref. [49] with permission from the American Chemical Society. Copyright © 2008, American Chemical Society).

At $S^1$ a water molecule reacts with the surface to form surface bound OH* (* represents the vacant site) at the CUS site:

$$2H_2O + * \xrightarrow{A} HO* + H_2O + H^+ + e^- \xrightarrow{B} O* + H_2O + 2H^+ + 2e^- \xrightarrow{C}$$
$$HOO* + 3H^+ + 3e^- \xrightarrow{D} O_2 + * + 4H^+ + 4e^- \tag{25}$$

At $S^2$ a water molecule reacts with O* to form OOH* and the following steps lead to the formation of $O_2$:

$$2H_2O + O* \xrightarrow{C} HOO* + H_2O + H^+ + e^- \xrightarrow{D} O_2 + H_2O + * + 2H^+ +$$
$$2e^- \xrightarrow{A} O_2 + HOO* + 3H^+ + 3e^- \xrightarrow{B} O_2 + O* + 4H^+ + 4e^- \tag{26}$$

The free energies of the different intermediates formation on surfaces $S^1$ and $S^2$ are shown in Figures 4 and 5, respectively, at pH = 0 and 14 with varying applied potentials (*U*). It was shown that each step goes downhill with the application of potential. The $S^2$ surface with total O covered is the most favored one at pH = 0 as it requires the lowest overpotential, 0.78 V (i.e., *V* = 1.23 + 0.78 = 2.01 V), among the studied surfaces, although it has higher overpotential than $RuO_2$ and $IrO_2$. The difference in the values of potentials for $S^1$ and $S^2$ are due to the different occupation of the nearest-neighbor CUSs. $TiO_2$ gain enough free energy via photo illumination to overcome the overpotential of water oxidation.

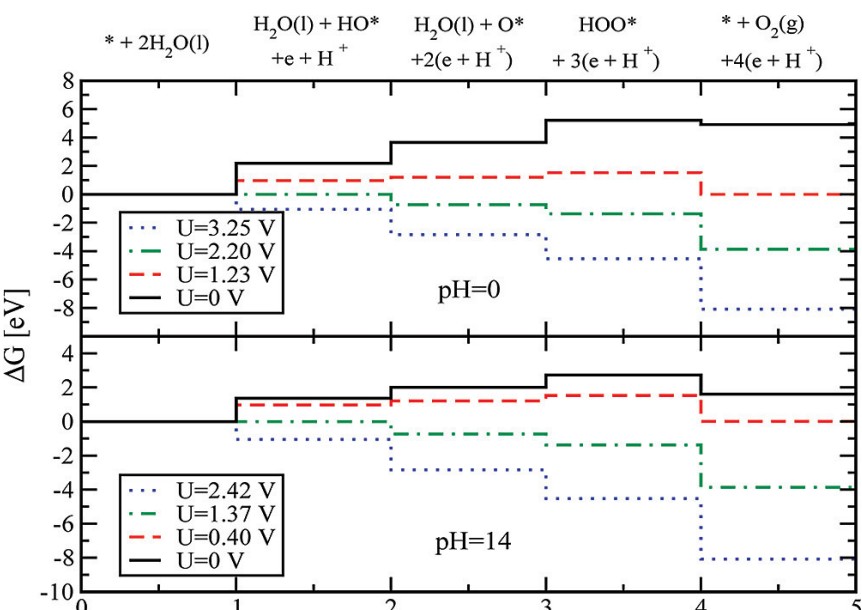

**Figure 4.** In the case of $S^1$ surface termination, the free energies of the intermediates at pH = 0(14) ($U$ = 0, 1.23(0.40), and 2.20(1.37) V) are shown. Some reaction steps are uphill in free energy at the equilibrium potential ($U$ = 1.23(0.40) V). All reaction steps are downhill in free energy at pH = 0(14). The corresponding values of the redox potential is 2.20(1.37) V. For the photogenerated holes the corresponding values of redox potential vs. NHE in the valence band $TiO_2$ (rutile) in contact with $H_2O$ under irradiation at pH = 0(14) are also compared. (Reprinted from ref. [49] with permission from the American Chemical Society. Copyright © 2008, American Chemical Society).

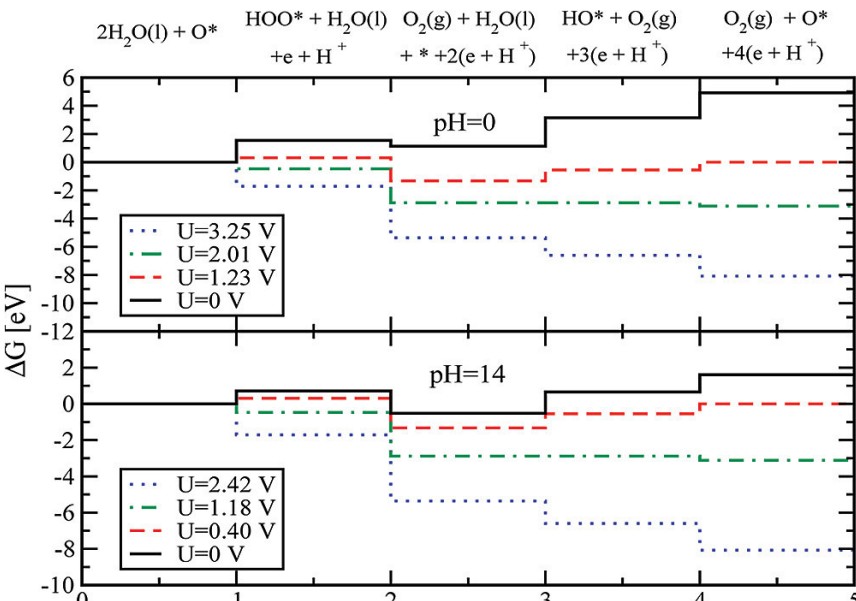

**Figure 5.** For the $S^2$ surface termination of $TiO_2$ the free energies of the intermediates at pH = 0(14) ($U$ = 0, 1.23(0.40), and 2.01(1.18) V) are presented. Some reaction steps are uphill in free energy at the equilibrium potential ($U$ = 1.23(0.40) V). All reaction steps are downhill in free energy at $U$ = 2.01(1.18) V and pH = 0/14. For the photogenerated holes, the corresponding values of redox potential vs. NHE in the valence band $TiO_2$ (rutile) in contact with $H_2O$ under irradiation at pH = 0(14) are also compared. (Reprinted from ref. [49] with permission from the American Chemical Society. Copyright © 2008, American Chemical Society).

Wang et al. [55] have investigated the photocatalytic OER mechanism using extensive first-principles molecular dynamics (MD) simulations at the water/rutile-TiO$_2$(110) interface. The surface-reaching photo-holes ($C_{h+}$) determine the rate limiting factors. It was shown that increased $C_{h+}$ enhances the OER activity. Two important reactions are the dissociation of adsorbed water to form surface active species and formation of the O-O bond to ultimately evolute O$_2$ gas. Three possible pathways (Figure 6) of the water dissociation were investigated: (i) adsorbed water (H$_2$O$_{ad}$) donates a proton to a bridged oxygen (O$_{br}^{2-}$) to form OH$_t^-$ and an O$_{br}$H$^-$, which is the most widely suggested pathway, (ii) Grotthuss-type proton transfer (H$_2$O$_{ad}$) directly release protons to the solution producing OH$_t^-$ and H$^+$(sol), and (iii) H$_2$O$_{ad}$ deprotonates in the presence of $C_{h+}$ generating OH$_t^-$ and H$^+$(sol). The activation barrier for the above-mentioned three processes are f 0.97 eV (TS9), 0.51 eV (TS1), and 0.39 eV (TS8), respectively. Thus, the presence of $C_{h+}$ promotes the water dissociation kinetically. The formation of radicals (OH$_t$ and O$_{br}^-$) in the presence of h$^+$ (OH$_t^- +$ h$^+ \rightarrow \cdot$OH$_t$ and O$_{br}^{2-} +$h$^+ \rightarrow$ O$_{br}\cdot^-$) is more favored than in absence. In the next step, to form the O-O bond, only the radical couplings are favored. The general mechanism of the reactions involving the Ti site and bridged oxygen sites are shown in Figure 7.

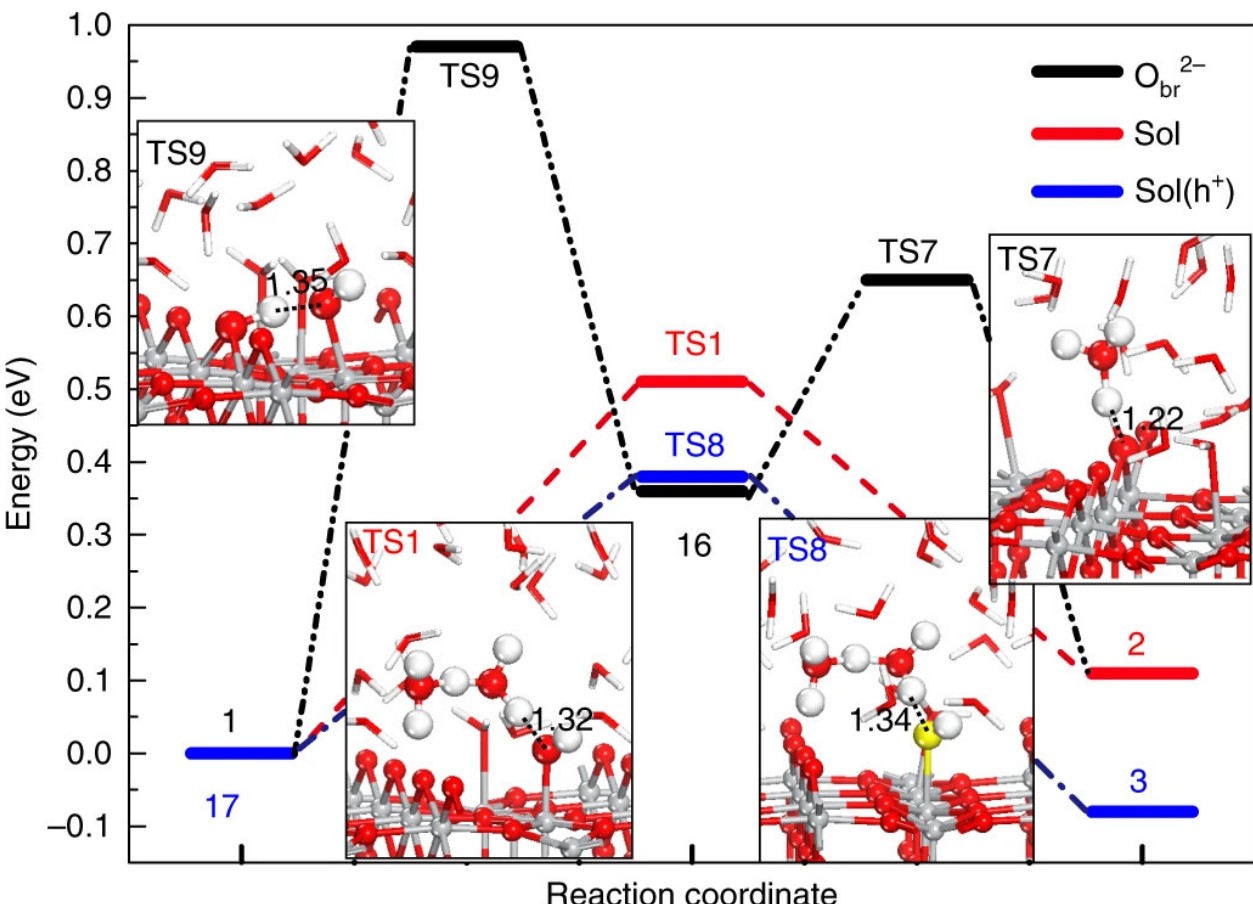

**Figure 6.** The energy profile diagram of the possible water splitting reaction. At the HSE06 level of theory, three possible pathways are considered: (i) first deprotonation to a nearby O$_{br}^{2-}$ and then into the solution (black), (ii) direct deprotonation into the solution (red), and (iii) with the assistance from the hole for the deprotonation (blue). In the insets the TS structures are shown and the surface radicals are highlighted in yellow (the lengths are in Å unit). (Reprinted from ref. [55] with permission from Nature Catalysis. Copyright © 2018, Nature).

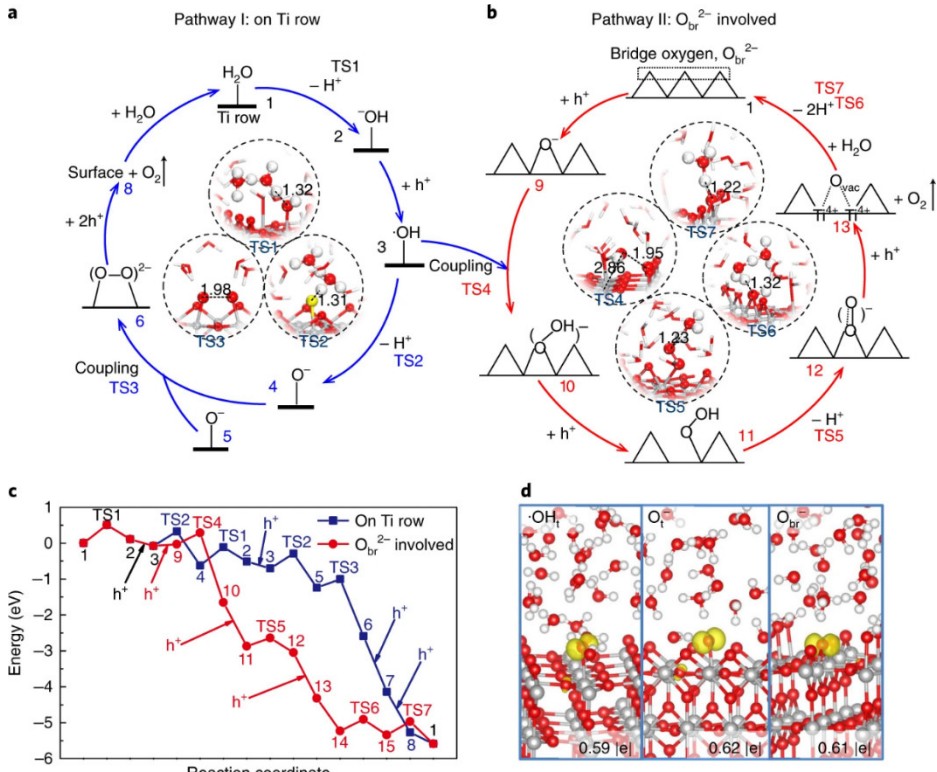

**Figure 7.** The photocatalytic energy profile diagram OER involving dual pathways of proposed mechanism. (**a**), Pathway I, the reaction pathway occurring on the Ti row. (**b**) Pathway II, the reaction involving bridge oxygen. The TS structures are shown in the insets., The states 1, 2, . . . , 15 correspond to the states in (**a**,**b**) in the energy profiles diagram of pathways I and II. (**c**) $h^+$ represents the involvement of holes in the elementary. (**d**) Spin-density plots showing the existence of three surface radicals (the O 2p orbital can be seen as a dumbbell shape with an iso-value of 0.005) and the Bader charge difference (~0.6 |e| signifies that after trapping $h^+$ the oxygen becomes +0.6 |e| positively charged with respect to the lattice oxygen of the bulk $TiO_2$. (Reprinted from ref. [55] with permission from Nature Catalysis. Copyright © 2018, Nature).

Li et al. [89] have utilized differently structured (101), (001), and (102) surfaces of anatase $TiO_2$ in aqueous medium to systematically compare the microscopic mechanisms of the oxygen evolution reaction (OER) by combining first-principles density functional theory calculations and a parallel periodic continuum solvation model. The high overpotential is dictated by the first proton removal. Only at an overpotential of 0.7 V (i.e., $V$ = 1.93 V vs. SHE) is the rate controlling surmountable at room temperature, the corresponding free energy change for the (101), (102), and (001) surfaces are 0.69, 0.63, and 0.61 eV, respectively. It was concluded that local surface structures of anatase have little effect on the OER mechanism. The presence of visible light and co-doping with high valent metals (Nb, Mo) enhance the OER.

The three polymorphs of $TiO_2$ are rutile, anatase, and brookite. These polymorphs show significant differences in their photocatalytic and photo-electrocatalytic water oxidation reactions, owing to the differences in their surfaces. In order to get better insight on the difference in the mechanism and energies of various processes, Li and co-workers [58] have studied it through density functional theory (DFT) on rutile $TiO_2$ (110), anatase $TiO_2$ (101), and brookite $TiO_2$ (210) model surfaces. The selectivity towards the formation of $^\bullet OH$, $H_2O_2$, or $O_2$ are studied through the construction of a potential-dependent free energy diagram in the water oxidation process. The following reactions are considered:

$$\text{Step 1: } * + H_2O \rightarrow OH^* + (H^+ + e^-) \tag{27}$$

$$\text{Step 2: OH}^* \rightarrow \text{O}^* + (\text{H}^+ + \text{e}^-) \tag{28}$$

$$\text{Step 3: O}^* + \text{H}_2\text{O} \rightarrow \text{OOH}^* + (\text{H}^+ + \text{e}^-) \tag{29}$$

$$\text{Step 4: }^*\text{OOH} \rightarrow \text{O}_2 + (\text{H}^+ + \text{e}^-) \tag{30}$$

where * represents the surface bound states. The free energies of one, two, and three electron processes are represented as $\Delta G_{\text{OH}^*}$, $\Delta G_{\text{O}^*}$, $\Delta G_{\text{OOH}^*}$, and $\Delta G_{\text{O2}}$ for the formation of OH*, O*, OOH*, and O$_2$. As one can see, all the above-mentioned steps are proton coupled electron transfer processes with an overall potential of 2.73, 1.76, and 1.23 V, respectively, for the one, two, and three electron oxidations. The structures of rutile (110), anatase (101), and brookite (210) are shown in Figure 8 with special emphasis on the position of OH*, O*, and OOH* intermediates on the rutile (110) surface.

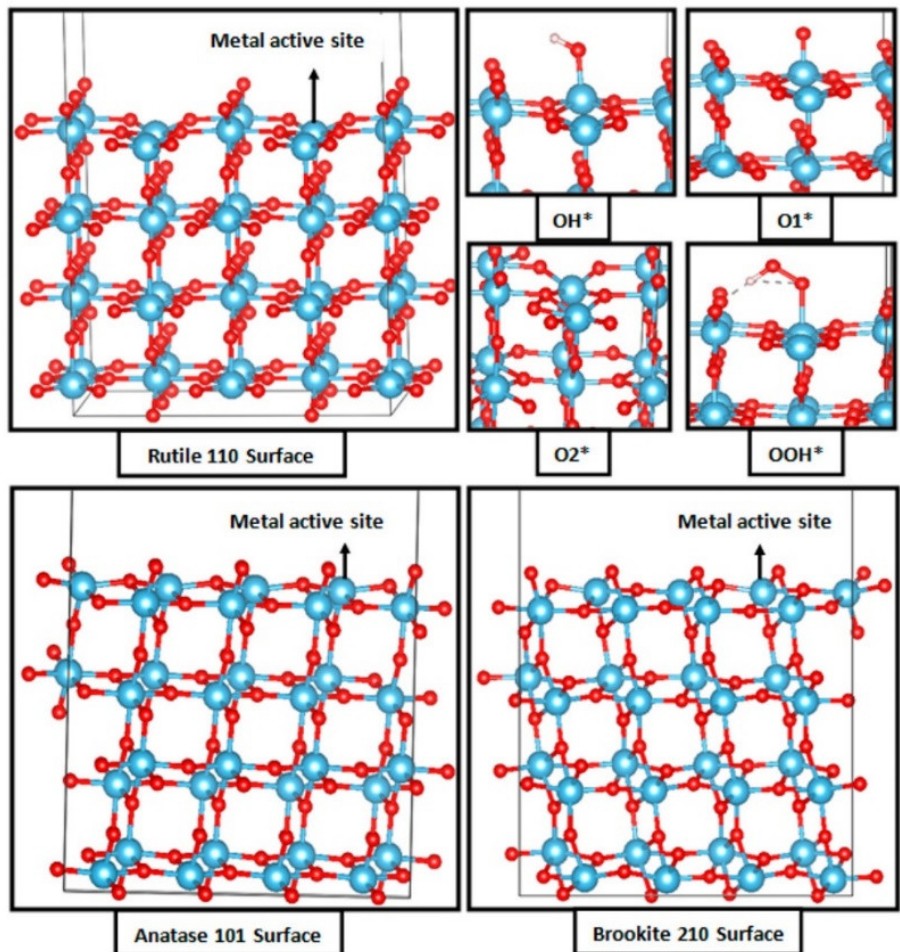

**Figure 8.** TiO$_2$ anatase (101), and brookite (210), and rutile (110), surfaces. The intermediate structures: dangling O*, OH*, OOH*, peroxo O* are shown in the upper right corner. Color code: Ti, light blue; O, red; H, white. (Reprinted from ref. [58] with permission from the American Chemical Society. Copyright © 2020, American Chemical Society).

The free energy diagram of water oxidation processes for TiO$_2$ rutile (110), anatase (101), and brookite (210) are shown in Figure 9. It was assumed that the reaction proceeds with the lowest free energy states without any knowledge of kinetic barriers among the intermediates. A dangling O* intermediate requires a higher free energy than that of surface peroxo O* intermediates and hence the latter is the mechanism. For comparison these values are (surface peroxo O* vs. dangling O*): rutile, ~1.66 eV vs. ~2.72 eV; anatase, ~1.64 eV vs. ~2.55 eV; and brookite, ~1.67 eV vs. ~2.53 eV.

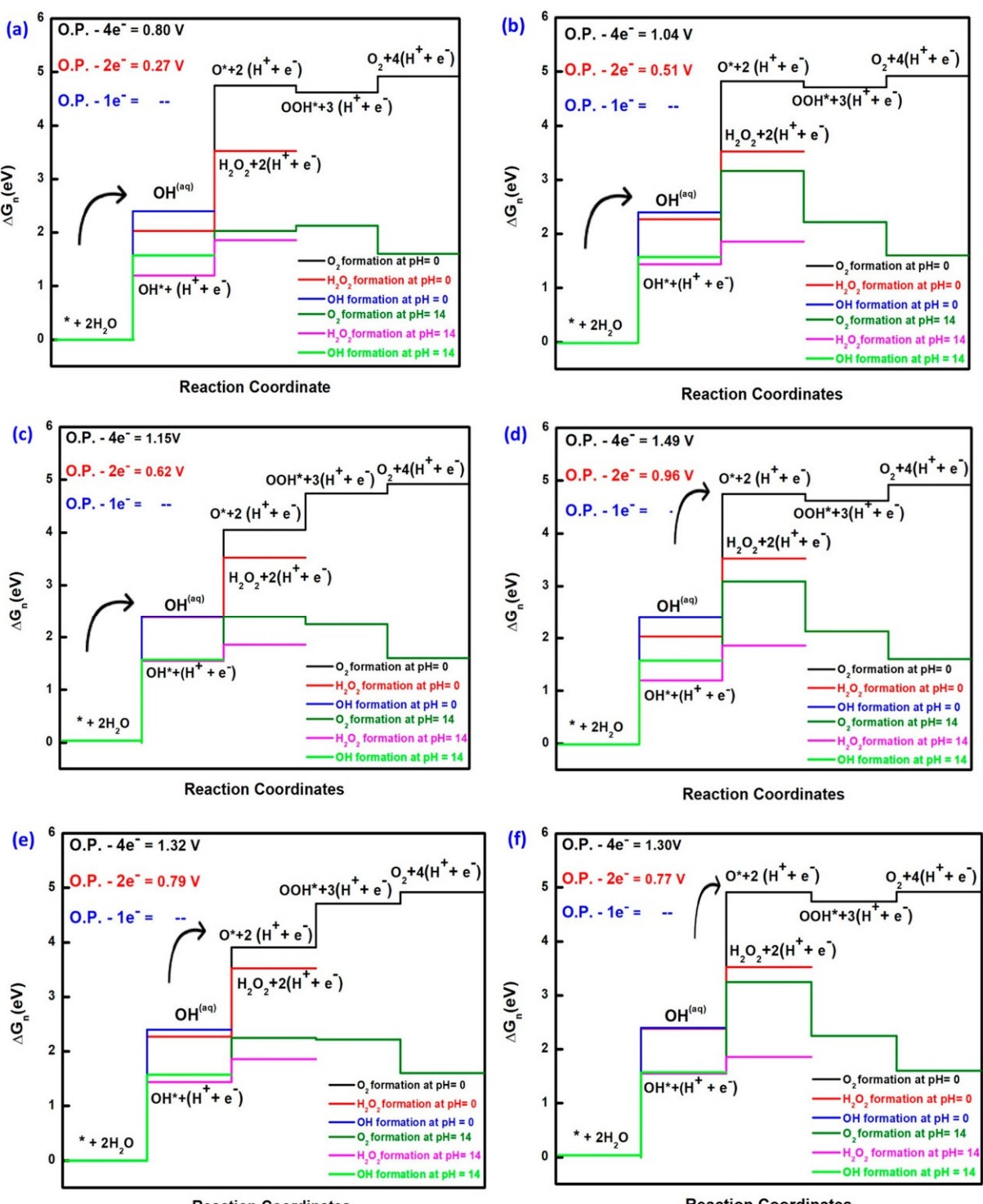

**Figure 9.** OER free energy diagrams involving peroxo O2* species on (**a**) TiO$_2$ rutile (110), (**b**) anatase (101), and (**c**) brookite (210) and through dangling O1* species on (**d**) TiO$_2$ rutile (110), (**e**) anatase (101), and (**f**) brookite (210) without any external potential. (Red line = generation of H$_2$O$_2$ at pH = 0; Black line = evolution of O$_2$ at pH = 0; blue line = generation of OH t pH = 0; pink line = generation of H$_2$O$_2$ at pH = 14; dark green profile = O$_2$ formation at pH = 14; light green line = generation of OH at pH = 14.) In all pathways the presence of peroxo O* intermediate can be seen. The arrow represents the rate-determining steps. In the left upper corner the values corresponding to 1e$^-$, 2e$^-$, and 4e$^-$ processes at pH = 0 are shown. (Reprinted from ref. [58] with permission from the American Chemical Society. Copyright © 2020, American Chemical Society).

Ferric oxide ($Fe_2O_3$) has been extensively studied as a photo-electrocatalyst over the past two decades [90–93]. In $Fe_2O_3$, solar energy absorption is feasible due to the suitable band gap and proper orientation of the band edge position. However, the most important aspect is its durability, which makes the hematite superior to other comparable catalysts [94]. The limitations are the high rate of recombination and low mobility [95–97]. It was suggested that these limitations could be overcome through nano-structuring [98–101].

Recently, Dahan et al. [102] studied the photoelectrochemical properties of fabricated nanoscale $Fe_2O_3$, which increases the solar energy absorption and slow diffusion length of charge carriers. It was shown that promoting hydrogen vacancy formation at high pH and voltage improves the catalytic activity. The effect of geometrical strain that originates due to the deposition on a surface was also discussed, and it was prescribed that the material should be deposited on materials of similar lattice constant to minimize lattice strain and increase efficiency.

One promising material for the photoelectrocatalytic water oxidation is $WO_3$. It has a narrow band gap of 2.6 eV compared to that of $TiO_2$; hence, a broad spectrum of visible light can be used [103]. Further, $WO_3$ shows high electrical conductivity, and recent studies suggest a higher activity for water oxidation compared to that of $TiO_2$, $Fe_2O_3$, and $BiVO_4$, respectively. The Hall mobility of $WO_3$ and rutile-$TiO_2$ are 12 and 0.3 $cm^2\,V^{-1}\,s^{-1}$, respectively. The theoretically determined photocurrent density of 1.5 is 5.0 $mAcm^{-2}$ at an AM of 100 $mWcm^{-2}$. This was attributed to the low energy valence band. $WO_3$ produces surface vacancies that trap electrons. The limitation of the material is that the conduction band is not suitable for $H_2$ reduction. Very recently, Teusch et al. [103] described the mechanism of water splitting using periodic boundary conditions.

The interaction of water with (001)-$WO_3$ was investigated, and hence the mechanism of water splitting was investigated [103]. Not only the intermediates but also the transition states of the reactions were studied. It was shown that the highest activation energy is required for the dissociation of water molecules into OH and H fragments. The water adsorption is favored at lower coverage, while the dissociated species are favored at higher coverage. It was shown that water splitting does not proceed without an external influence.

### 3.2. Manganese Oxide-Based Catalyst

In the photosynthesis process the photosystem II contains the oxygen evolving complex (OEC) $CaMn_4O_5$ (Figure 10), which efficiently oxidizes water at an overpotential of ($\eta$) of 160 mV at pH 6.5 and with a higher turnover frequency (TOF of 100–400 $s^{-1}$) than the most promising artificial water oxidation catalysts [104–106]. Considering the role of manganese in biological water oxidation, it is highly promising to develop a manganese oxide-based water oxidation catalyst. Manganese oxide exhibits over 30 different crystalline structures in nature [107,108]. From oxidation states $Mn^{II}$ to $Mn^{IV}$ some oxides of Mn are MnO, $Mn_3O_4$, $Mn_2O_3$, $Mn_5O_8$, and $MnO_2$, having stoichiometric chemical formulas. $Mn_2O_3$ consists of five symmetry-inequivalent $Mn^{III}O_6$ octahedra connected by corner and edge. Meanwhile, $Mn_3O_4$ is a mixed-valence manganese oxide having $Mn^{II}O_4$ tetrahedra and Jahn–Teller distorted $Mn^{III}O_6$ octahedra. $MnO_2$ can have one-dimensional $MnO_6$ octahedra chain (e.g., $\alpha$-, $\beta$-, $\gamma$-, and R-$MnO_2$), two-dimensional sheets of $MnO_6$ octahedra (e.g., $\delta$-$MnO_2$), and three-dimensional structures (e.g., $\lambda$-$MnO_2$) [109]. The $\beta$-$MnO_2$ structure is thermodynamically stable over all pH ranges. In manganese, oxide, and hydroxide it exists over a wide pH range [1]. Different oxide structures exhibit distinct electrochemical behavior [110]. The flexible Mn-O moieties are beneficial to mimic the biological WOC. In this regard, a variety of Mn-based oxides have been synthesized and employed as a catalyst for the water oxidation reaction [111–113]. This section outlines the theoretical understanding of mechanism of OER using manganese oxide materials.

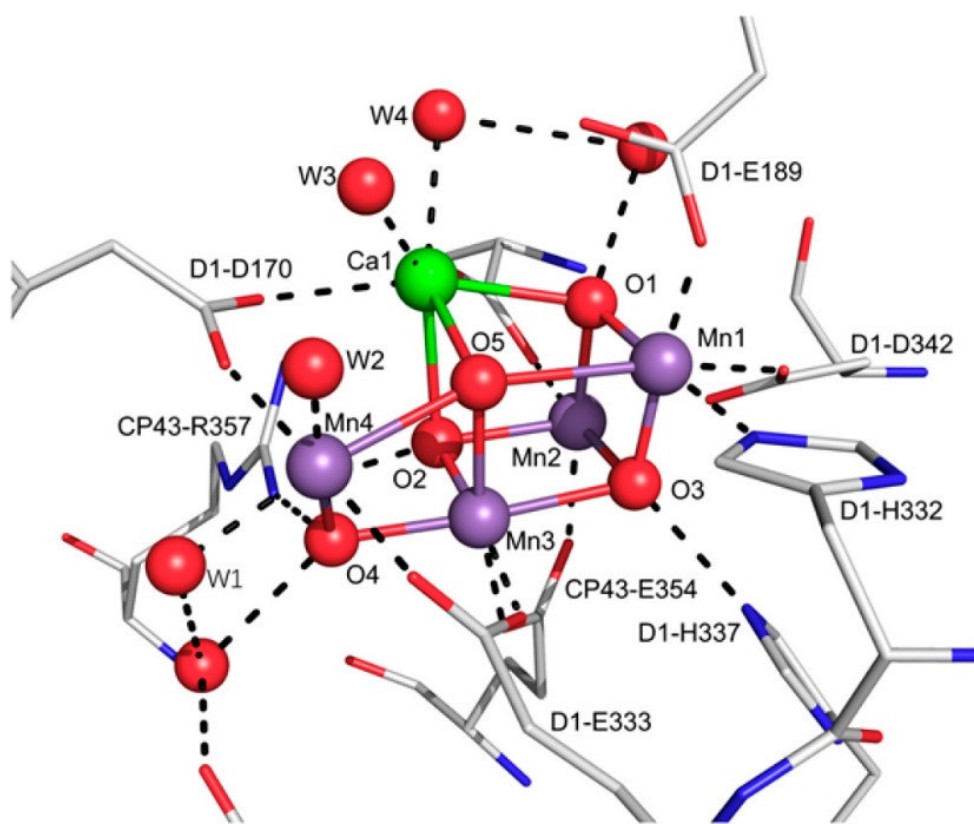

**Figure 10.** The active site structure of PS II. (Reprinted with permission from ref. [105] Copyright 2011 Nature Publishing Group and Wikipedia). Color code: C, grey; N, blue; O, white; Ca, green; Mn, violet.

Thus, manganese oxide $MnO_x$ serves as a promising material for the OER [106]. The commonly known polymorphs of $MnO_2$; α- (tetragonal), β- (tetragonal), and δ-(triclinic), are poor in their OER activity with a high overpotential (>0.6 V at pH < 7) [8,114]. In contrary, the electrochemically synthesized induced layer δ-$MnO_2$ (EI-δ-$MnO_2$) shows comparable performance to precious metal oxides (Ir/$RuO_x$) at an overpotential of ∼0.54 V at pH 0.3 [8,114–116]. Since these EI-δ-$MnO_2$ are dynamically produced in situ from spinel $Mn_3O_4$ (Figure 11) under electrochemical OER condition, their structures are hence uncertain on an atomic level.

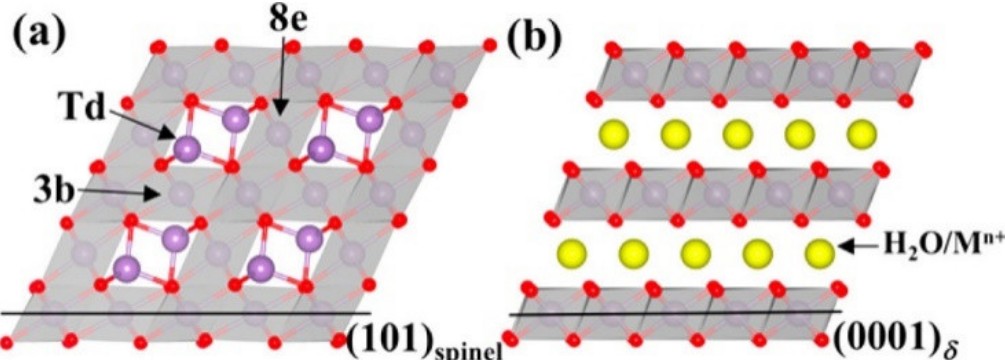

**Figure 11.** The polyhedral crystal structures representations of (**a**) spinel $Mn_3O_4$ (#141, *I41/AMD*) and (**b**) layered δ-$MnO_2$ (#166, *R3̄m*). Color code: O, red; H, white, Mn, violet. The yellow sphere represents the presence of $H_2O/M^{n+}$. (Reprinted from ref. [60] with permission from American Chemical Society. Copyright © 2018, American Chemical Society).

Li et al. [60] have demonstrated that the EI-δ-MnO$_2$ with a special edge site with neighboring Mn vacancy provides excellent OER activity using first-principles calculations with an overpotential of 0.59 V, 0.19 V lower than that of pristine MnO$_2$. As a first step, Li et al. [60] have utilized the thermodynamics approach to analyze the activity for OER on pristine spinel Mn$_3$O$_4$ (001) and δ-MnO$_2$ (2116) (Figure 12). In their study, they first studied the mechanism of OER on the pristine spinel Mn$_3$O$_4$ and δ-MnO$_2$ to compare the activity of in situ generated poorly crystallized EI-δ-MnO$_2$. Two possible mechanisms: (a) H$_2$O → OH → O → O$_2$ (31–34) and (b) H$_2$O → OH → OOH → O$_2$ (Reactions (31), (32), (35), and (36)) were considered.

$$2*H_2O \rightarrow *OH + *H_2O + H^+ + e^- \tag{31}$$

$$*OH + *H_2O \rightarrow 2*OH + H+ + e- \tag{32}$$

$$2*OH + *H_2O \rightarrow *OH* + *O + H^+ + e^- \tag{33}$$

$$*OH* + *O \rightarrow O_2(g) + H^+ + e^- \tag{34}$$

or

$$*OH \rightarrow *OOH + H^+ + e^- \tag{35}$$

$$*OOH \rightarrow * + O_2 + H^+ + e^- \tag{36}$$

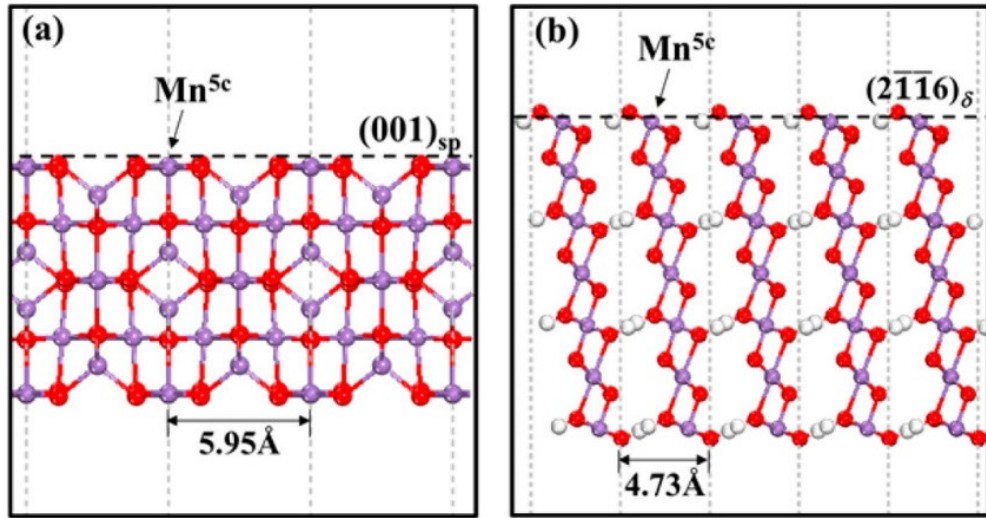

**Figure 12.** Representation of surface structures for (**a**) (001)$_{sp}$ and (**b**) (2116)$_δ$ of Mn$_3$O$_4$ (001) and δ-MnO$_2$ (2116), respectively. Color code: O, red; H, white, Mn, violet. (Reprinted from ref. [60] with permission from American Chemical Society. Copyright © 2018, American Chemical Society).

In the formula, the asterisk * denotes the edge Mn$^{5c}$ sites of layered MnO$_2$. The surface covered with two water molecules sequentially remove protons to the aqueous solution to form one monolayer (ML) OH covered surface (2*OH, Figure 13). The *OH further releases a proton to form *O, which combines with a lattice O to form O–O peroxo species (*OH + *O, Figure 13). The last proton is released along with the simultaneous evolution of oxygen. Alternatively, the *OOH may be formed by the combination of surface adsorbed *OH. The intermediate states and the energy profile diagram are shown in Figure 13. For the pristine spinel Mn$_3$O$_4$ (001) and δ-MnO$_2$ (2116), the lowest energy pathway follows the H$_2$O → OH → O → O$_2$ (31–34) mechanism.

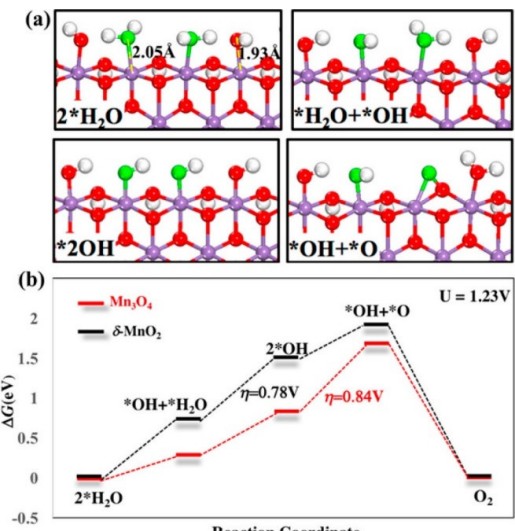

**Figure 13.** (**a**) The intermediates of the OER on the edge site of δ-MnO$_2$. H$_2$O and terminal OH become adsorbed on the Mn$^{5c}$. The structures of intermediates on spinel Mn$_3$O$_4$ and δ-MnO$_2$ of OER are similar. (**b**) OER energetic profiles diagram on Mn$_3$O$_4$ and δ-MnO$_2$. Color code: O, red; H, white, Mn, violet. The oxygen that are involved in the reaction are represented by green color. (Reprinted from ref. [60] with permission from American Chemical Society. Copyright © 2018, American Chemical Society).

The phase diagram of the transformation of Mn$_3$O$_4$ to H$_x$MnO$_2$ and to MnO$_2$ at pH = 7 is shown in Figure 14 (the relevant equations are shown in Equations (37)–(39)).

$$2Mn_3O_4 + 4H_2O \rightarrow 6H_{0.5}MnO_2 + 5H^+ + 5e^- \tag{37}$$

$$4H_{0.5}MnO_2 \rightarrow 4H_{0.25}MnO_2 + H^+ + e^- \tag{38}$$

$$4H_{0.25}MnO_2 \rightarrow 4MnO_2 + H^+ + e^- \tag{39}$$

It was shown that at potentials < 1 V, the pristine Mn$_3$O$_4$ is stable, corroborating the experimental results. The intermediate phase H$_{0.5}$MnO$_2$ becomes prevalent above 1 V. With increasing potential, H$_{0.5}$MnO$_2$ gradually loses a proton to ultimately form MnO$_2$ at 1.35 V. Experimentally, Kim et al. [117] have observed that the spinel-to-layer phase transition occurs in the potential range 0.9–1.1 V.

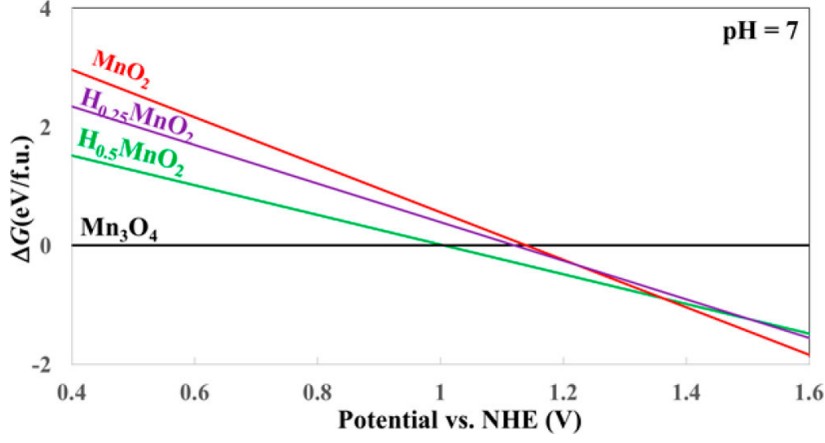

**Figure 14.** Calculated phase diagram for spinel Mn$_3$O$_4$, H$_{0.5}$MnO$_2$, H$_{0.25}$MnO$_2$, and MnO$_2$ at pH 7. (Reprinted from ref. [60] with permission from American Chemical Society. Copyright © 2018, American Chemical Society).

The kinetic energy barrier for the spinel-to-layer transition is lower in the presence of proton. The whole transition has an overall barrier of 0.35 eV/f.u. and the rate determining step is related to Mn ions embedding into the Mn vacancy of the $MnO_2$ layer. In the absence of proton, the transition energy barrier is 0.49 eV and hence kinetically becomes less favorable. Two sequential layering movements of $MnO_2$ layers along with [$1\bar{2}10$] and [$11\bar{2}0$] and simultaneous $Mn^{ic}$ diffusion are the pathways of spinel-to-layer phase transition. The nascent δ-$MnO_2$ layers peel off from the spinel matrix and further break apart to form new $(0001)_δ$/[$11\bar{2}0$] edges due to the generation of strain between spinel and δ-$MnO_2$.

The three different local structures, adatom, dislocation, and edge-NV, as obtained in the solid phase transition mechanism, were examined during OER activity. Only the edge-NV site shows promising OER activity, where the overpotential is 0.19 V lower compared to the pristine edge site. The edge-NV site has a Mn vacant site and mixture of $Mn^{3+}$ and $Mn^{4+}$ cations, as a result the local $Mn^{4+}$ concentration is increased near the OER site. The high activity of the edge-NV site is ascribed to the cubane local structure with a missing corner, which promotes the intermediate OH adsorption. Thus, the enhanced OER activity of electrochemical synthesized nascent EI-δ-$MnO_2$ is attributed to the presence of edge-NV sites.

A lot of experimental work has been performed to understand the electrochemical and photoelectrochemical OER activity of $MnO_2$ polymorphs [118–129]. Gupta et al. [130] have shown the enhanced electrocatalytic activity of the OER through stabilization of metastable or non-native (NN) polymorphs of $MnO_2$. They have synthesized β/N-, γ/N-NN1-, r/NN1-, α/NN2-, and δ/NN3-$MnO_2$ polymorphs of $MnO_2$, which show different OER activity. Using DFT descriptor $\Delta G_{O*} - \Delta G_{HO*}$, a volcano-based relationship of $MnO_2$ polymorphs was observed, where δ/NN3-$MnO_2$ and α/NN2-$MnO_2$ lie closer to the volcano peak. The higher specific activity of δ/NN3-$MnO_2$ is attributed to the low oxidation state (+3.5) of Mn as obtained from the X-ray photoelectron spectroscopy (XPS), which gives average oxidation states (AOS) and Bader charge analysis. α/NN2-$MnO_2$ requires less energy for the oxygen vacancy formation and hence shows higher activity. Another reason of high activity in δ/NN3-$MnO_2$ and α/NN2-$MnO_2$ are stronger O adsorption attributed to the shift in Mn-d to the Fermi level.

The OER activity was explained through the band structure analysis (Figure 15). The OER activity is also dependent on the strength of oxygen adsorption and coverage on the $MnO_2$ surface [131]. The bonding and antibonding states are formed as the coordinatively unsaturated Mn atom donates electrons to the adsorbed oxygen. The unoccupancy of the antibonding states depends on the valency of the Mn-d band. A high energy Mn-d band leads to a lesser occupancy and hence stronger adsorption of O. With decreased thermodynamic stability, the polymorphs become more non-native (NN) and the Mn-d band shifts upward to the Fermi, giving more O adsorption. Thus, δ/NN3-$MnO_2$ and α/NN2-$MnO_2$ resides at the top of the volcano in the specific OER activity.

Recently Zhou et al. [64] have synthesized the α- and δ-$MnO_2$, by changing the hydrothermal conditions of the synthesis. The α phase $MnO_2$ exhibits superior OER catalytic performance with an overpotential of 0.45 V. The metal oxides with smaller band gap shows higher electrical conductivity and better OER properties through the transfer of electrons [132,133]. The first principle based DFT study shows that the δ form has 1 eV higher band gap compared to that of α-$MnO_2$. They also calculated the alignment of the valence band (VB) and conduction band (CB) with respect to the vacuum level ($E_V$) through the calculation of $\Delta E_{VB} = E_V - E_{VB}$ and $\Delta E_{CB} = E_V - E_{CB}$. ($E_{VB}$ = energy of the valence band, and $E_{CB}$ = energy of the conduction band) For α-$MnO_2$ these values are 6.62 eV and 8.39 eV while they are 6.44 eV and 9.11 eV for δ-$MnO_2$ (Figure 16).

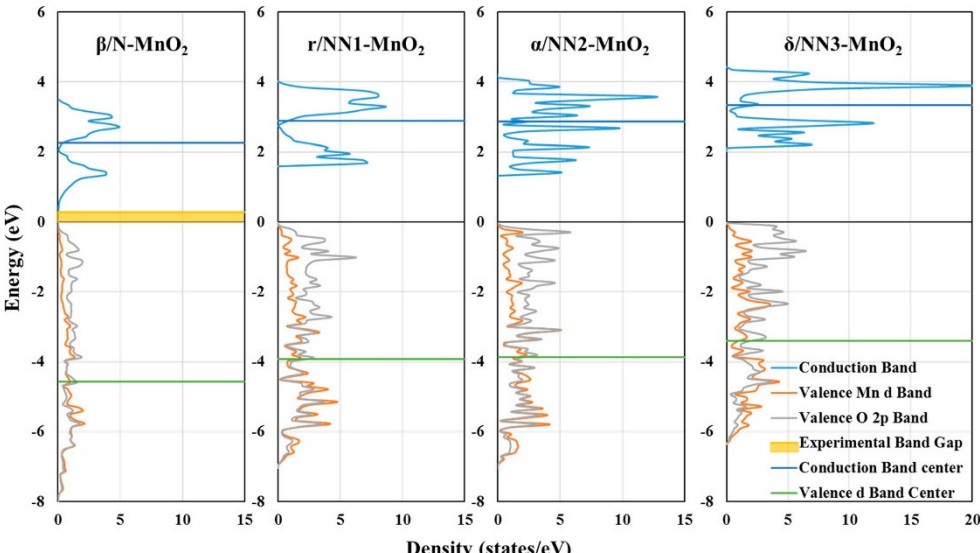

**Figure 15.** Density of states plot for different polymorphs of MnO$_2$. Valence Mn-d band center shifts upward and closer to the Fermi level as we move toward more non-native polymorphs. (Reprinted from ref. [130] with permission from the Journal of Physical Chemistry C. Copyright © 2019, American Chemical Society).

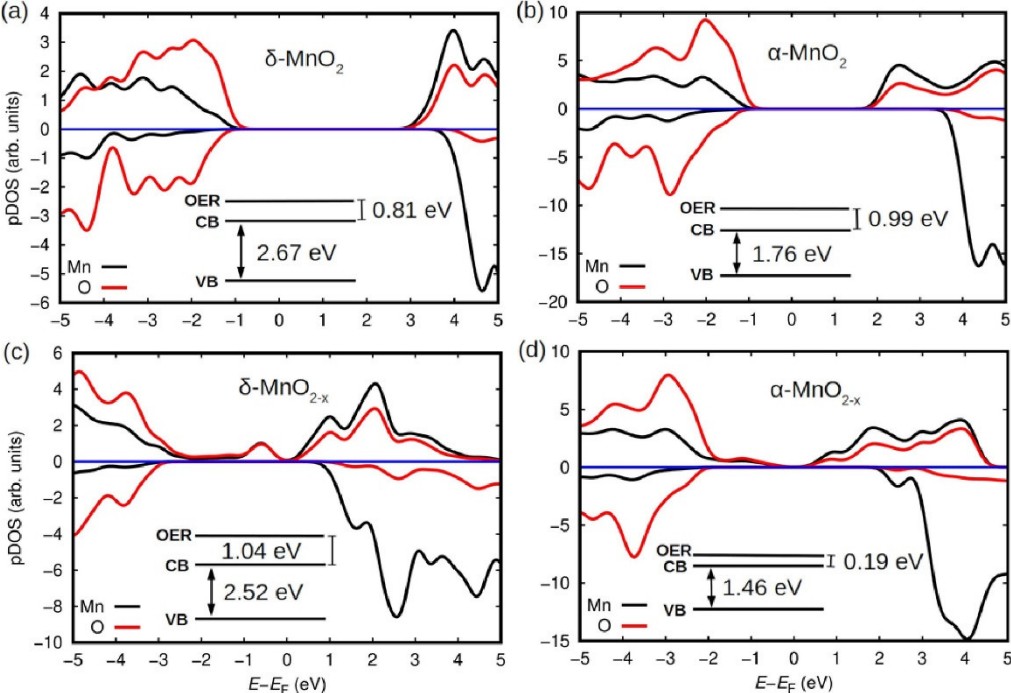

**Figure 16.** Theoretical estimation of the electronic band structure properties of (**a**) δ- and (**b**) α-MnO$_2$ and of (**c**) δ- and (**d**) α-MnO$_{2-x}$ based on first-principle DFT calculations. The spin-up (spin-down) electronic density of states is represented with positive (negative) values. "OER" stands for the energy level of the oxygen evolution reaction (O$_2$/H$_2$O), "CB" for the alignment of the conduction band, and "VB" for the alignment of the valence band. (Reprinted from ref. [64] with permission from the Chemical Engineering Journal. Copyright © 2022, Elsevier).

Selvakumar et al. [134] have shown by DFT that the 310 surface has an affinity towards water molecules and proposed that a needle type morphology is responsible for the higher exposure of such a surface to the OER reaction interface. The hollandite crystal structure of α-MnO$_2$ shows better catalytic activity compared to the other polymorphs.

Furthermore, comparative investigations consistently indicate that the hollandite crystal structure of $\alpha$-MnO$_2$ (Figure 17) delivers substantially better catalytic performance than other polymorphs of MnO$_2$. Here the oxygen defect formation plays a significant role in this regard. [135] The following equation was used to calculate the vacancy formation energy:

$$O_o^x + 2Mn_{Mn}^x \rightarrow V_{\ddot{o}} + 2Mn_{Mn}^x \tag{40}$$

where notations of Kroger-Vink were used.

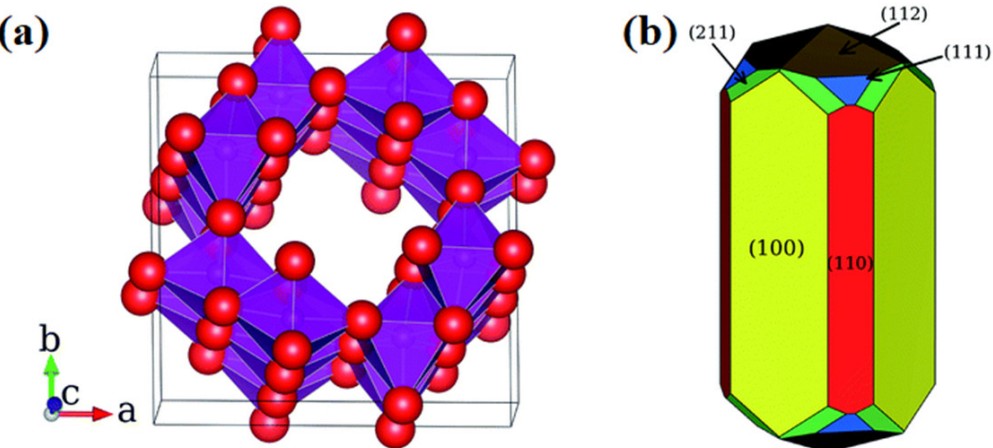

**Figure 17.** (**a**) Crystal structure of $\alpha$-MnO$_2$. Small (red) spheres are oxygen and large (purple) manganese lie inside the indicated approximate MnO$_6$ octahedra. Note the large 2 × 2 tunnel structures along the *c*-axis. (**b**) The predicted equilibrium crystal morphology for $\alpha$-MnO$_2$ based upon the surface energies. (Reprinted from ref. [135] with permission from the Journal of Materials Chemistry A. Copyright © 2014, Royal Society of Chemistry).

All surfaces have relatively low vacancy formation energy within the range of 0.07 eV to 1.09 eV. PBE + U surface energies for $\alpha$-MnO$_2$ were calculated using the following equation:

$$\gamma = \frac{E_s - nE_b}{2A} \tag{41}$$

The (100) and (110) surfaces, with low energies of 0.64 and 0.75 J·m$^{-2}$, respectively, are the most stable. All other surface energies are greater than 1 J·m$^{-2}$. Further the long side of the crystal contains the low-indexed 100 and 110 surfaces while the high-indexed occupy the needle tips.

Su et al. [136] demonstrated the pH dependence of MnO$_x$ bulk and surface structures and their effect on OER. They have considered three different bulk oxide phases: (a) Mn$_3$O$_4$(001), (b) Mn$_2$O$_3$(110), and (c) MnO$_2$(110) to investigate the relative stability of different adsorbate surface structures. The most likely surface is the stable one with the thermodynamically lowest free energy at a given potential. The general Pourbaix diagrams showing the changes in adsorbate surface structure as a function of pH and potential for three different MnO$_x$ bulk structures are shown in Figure 18, which account for phase transitions in both the bulk (e.g., in the near-surface region) and at the surface of the catalyst. The surface changes with increasing potential are the following: (i) 0.46 V < $U_{RHE}$ < 0.69 V, adsorbate-free Mn$_3$O$_4$(001) surface; (ii) 0.69 V < $U_{RHE}$ < 0.98 V, 1/2 ML HO* covered Mn$_2$O$_3$(110); (iii) 0.98 V < $U_{RHE}$ < 1.01 V, Mn$_2$O$_3$(110), and HO* coverage increases to 3/4 ML; and (iv) 1.01 V < $U_{RHE}$ < 1.21 V, Mn$_2$O$_3$ oxidized to MnO$_2$(110) and, at the surface, bridge sites are covered with HO*. Above 1.21 V the surface is completely covered by O* and even higher potential leads to dissolution in the form of MnO$_4^-$.

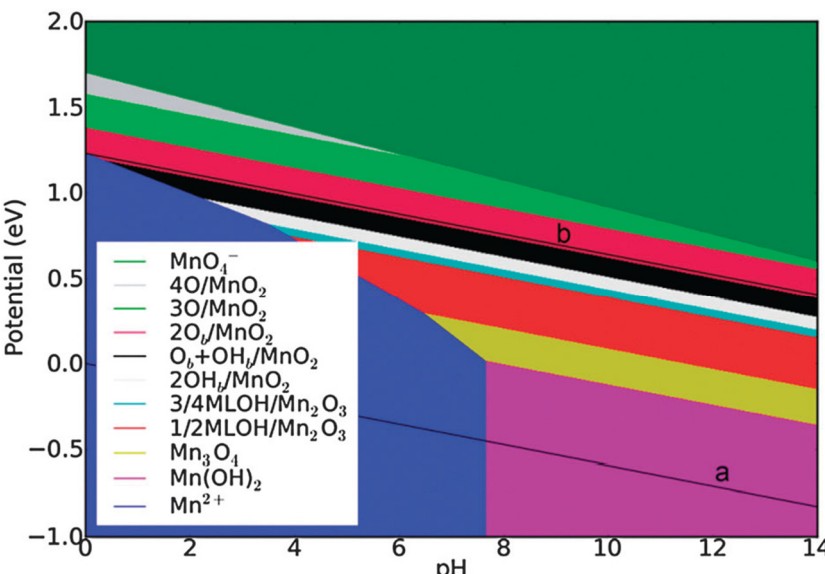

**Figure 18.** General surface Pourbaix diagram for $MnO_x$ catalysts. The oxidation state of the surface and the ORR and OER potential are constant versus the reversible hydrogen electrode (RHE). Lines a and b represent the RHE line and the $O_2/H_2O$ equilibrium line. (Reprinted from ref. [136] with permission from the Physical Chemistry Chemical Physics. Copyright © 2012, Royal Society of Chemistry).

The free energy diagram for the OER show that an associative pathway is energetically favorable in O* covered $Mn_3O_4(001)$ and O* covered $Mn_2O_3(110)$ compared to the direct pathway. However, in case of O* covered $MnO_2(110)$ surface the direct pathway is slightly favored. The intermediates have a better compromise in interaction strength as O* covered $MnO_2(110)$ surface is close to the top of the volcano [2]. In this case, the direct mechanism of recombination of oxygen atoms has a slightly lower free energy than the associative mechanism by only 0.08 eV. In $MnO_x$ the scaling relationship between HOO* and HO* holds as well with values of 3.18 eV, 3.1 eV, and 3.12 eV [136]. Due to the coverage of the adsorbate, slight deviation of $(\Delta G_{HOO} - \Delta G_{HO})$ from the ideal value of 3.2 is observed.

Rong et al. [137] have predicted the surface structure and stoichiometry in aqueous solutions of $LaMnO_3$ using first-principles computations and electrochemical principles. To predict accurate stable oxide surfaces, it is important to calculate the free energy changes associated with the formation of aqueous cation species as a function of $U$ and pH. On the perovskite type materials, two types of OER mechanisms are shown: (a) adsorbate–evolution mechanism (AEM) and (b) Mars–van Krevelen type mechanism (MKM) [2,138,139]. Through the AEM process the *OH, *O, and *OOH intermediates are formed at one site and ultimately $O_2$ is involved, which is relatively common. On the other hand, the lattice oxygen participates in the process of oxygen evolution and sometimes two sites act cooperatively to form a single $O_2$ molecule. However, in solution the perovskite materials undergo reconstruction and hence multiple local sites may form [137]. Thus, it is important to study the reactions on reconstructed surface of perovskite material.

Qiu et al. [140] have studied the catalytic activity for the OER on the perovskite-type material $CaMnO_3$. The lowering of OER overpotential is due to the Mn vacancy formation on the surface that reduces the energy difference of *OOH and *OH. The electrode potential for the OER on the $Mn_2O_4 - 1.0Mn - 1.0O$ surface is $U_{RHE} = 1.62$ V, which is consistent with the experimental value $U_{RHE} = 1.6$ V and the stability region of metal ion exchange and surface hydration and (de)hydrogenation. On the other hand, the electrode potential on the $Mn_2O_4 + 2.0O$ surface is $U_{RHE} = 1.90$ V which is ruled out as it is out of its ion exchange stability region. Thus, oxygen is evolved from the Mn site of the $Mn_2O_4 - 1.0Mn - 1.0O$ surface. The OER mechanism is shown in Figure 19. The whole process is classified as the MKM process as oxygen is involved though the involvement of lattice oxygen.

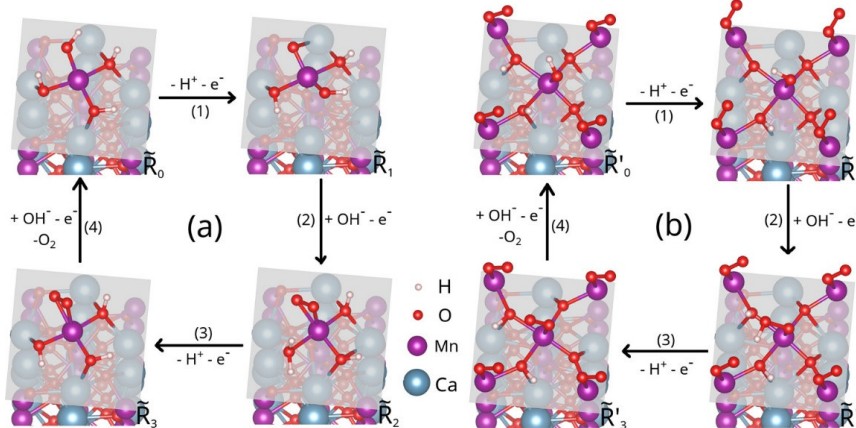

**Figure 19.** (**a**) OER cycle on surface $Mn_2O_4 - 1.0Mn - 1.0O$. (**b**) OER cycle on surface $Mn_2O_4 + 2.0O$. Surfaces are named with the format "termination ± #atom", where + and − represent adatoms and vacancies, respectively, and # represents the number of atoms introduced/removed per supercell relative to a reference surface with fixed termination, which can be either $Mn_2O_4$ or $Ca_2O_2$. (Reprinted from ref. [140] with permission from ACS Catalysis. Copyright © 2018, American Chemical Society).

Du et al. [141] have shown that non-stoichiometric $CaMnO_{3-\delta}$ ($\delta \approx 0.25$) with moderate oxygen-defects improves the electrical conductivity and hence the OER activity. Density of states (DOS) calculations were performed on stoichiometric $CaMnO_3$ (without oxygen vacancies), $CaMnO_{2.75}$ (one oxygen vacancy per unit cell), and $CaMnO_{2.5}$ (two oxygen vacancies per unit cell). The semiconducting nature of $CaMnO_3$ is attributed to the band gap ($E_g$) of approximately 1.56 eV. The DOS profile near the Fermi level changes significantly. There is a gap for the spin-down states while the spin-up states exist, which implies the half metallicity and hence increased electrical conductivity due to spin polarization. Despite spin polarization in $CaMnO_{2.5}$, no obvious states were found near the Fermi level and hence half metallicity is insignificant. The $O_2$ adsorbed structures on $CaMnO_3$ and $CaMnO_{2.75}$ surfaces are shown in Figure 20. The O-O rupture is favorable on $CaMnO_{2.75}$ surfaces as the $O_2$ is bonded to Mn atoms through a Griffith (side-on) manner. The O-O bond length is shorter (1.40 Å) in the $CaMnO_3$ surface than in the $CaMnO_{2.75}$ surfaces. This implies that oxygen vacant $CaMnO_{2.75}$ is more reactive compared to that of $CaMnO_3$.

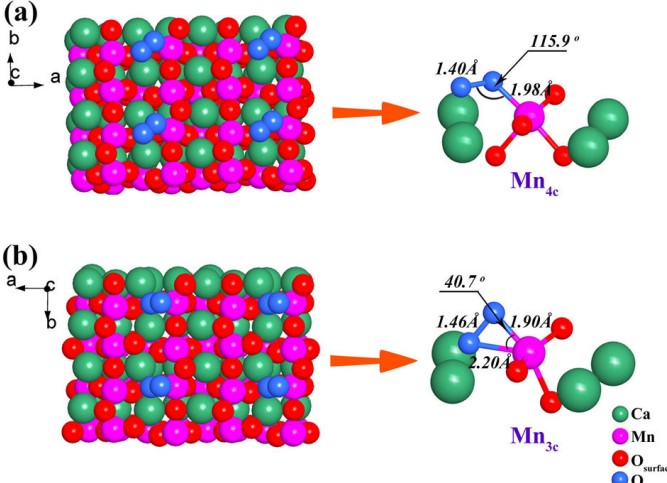

**Figure 20.** Top view of the configuration of molecular oxygen adsorbed on the (**a**) $CaMnO_3$ (121) surface and (**b**) $CaMnO_{2.75}$ (121) surface. The right parts are the bond length information after oxygen is adsorbed on the surfaces. The green, pink, and red spheres are used to represent Ca, Mn, and lattice O atoms, respectively, and the adsorbed oxygen molecule is shown in blue. (Reprinted from ref. [141] with permission from Inorganic Chemistry. Copyright © 2014, American Chemical Society).

### 3.3. Cobalt Oxide-Based Catalyst

Cobalt-based materials have gained significant attention among all other oxides due to their promising industrial application [142–144]. Nocera and co-workers reported on cobalt phosphate (Co–Pi) material in 2008. The Co-Pi catalyst was generated in situ by electrochemical deposition on an ITO film from a phosphate buffer solution containing $Co^{2+}$ [145–147]. The catalyst is active for OER under neutral pH and moderate overpotential. Similar studies have also been performed in cobalt carbonate and borate catalysts [148–150]. A vast amount of heterogeneous water oxidation reactions have been performed using the cobalt oxide material as an electro, photo, and photoelectron catalyst [151–155]. The recent development in the theoretical aspect of the mechanism of cobalt oxide-based materials is described here.

Mattioli et al. [65] have taken a cobalt oxide-based catalyst (CoCat) to perform DFT+U molecular dynamics calculations, including an explicit water solution to provide insight into the OER mechanism (Figure 21). They suggested that the cubane-like Co-oxo units' species are formed at the surface of the catalyst. They identified the formation of Co(IV)-oxyl species as the driving ingredient of the geminal coupling with O atoms coordinated by the same Co. A high activation barrier disregards the water nucleophilic attack (Figure 21). The CoCat shows similarity to the $CaMn_4O_5$-based PSII complex.

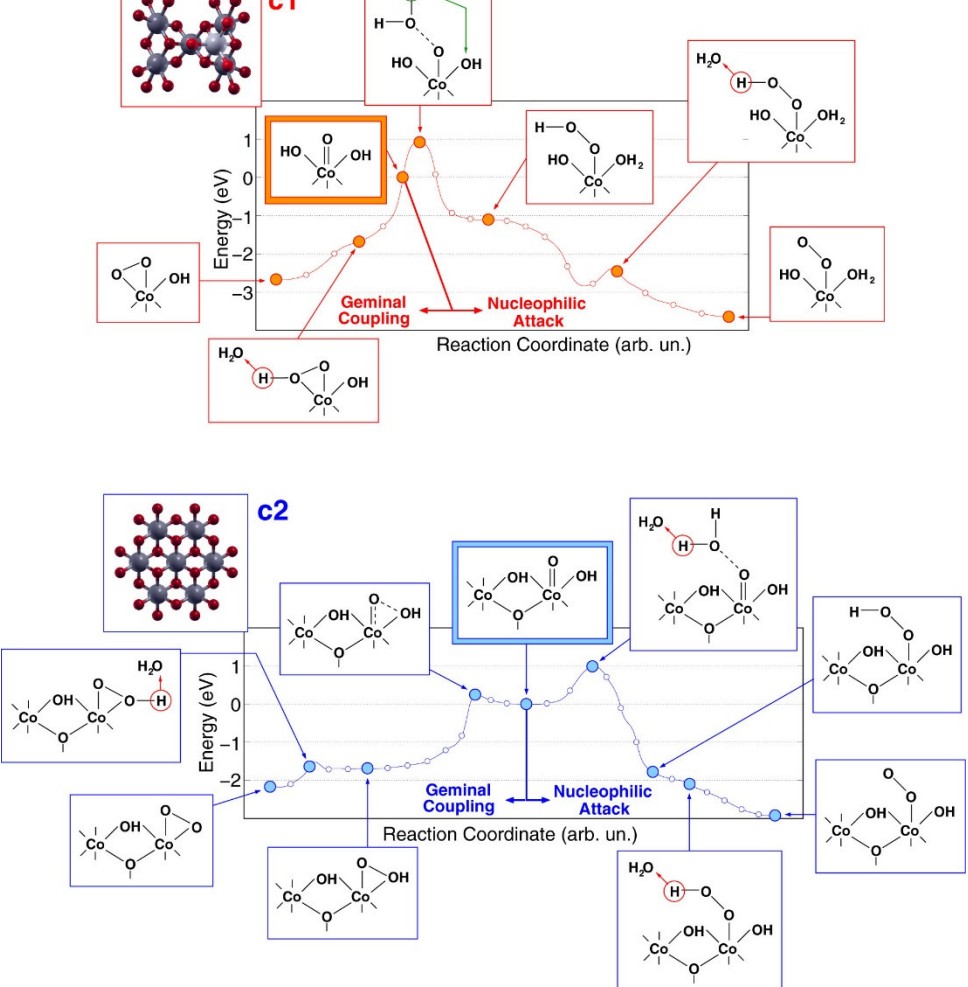

**Figure 21.** Reaction paths for oxygen evolution promoted by the c1 and c2 CoCat models. Details of the path A, including Co–O and O–O distances and spin-switching effects related to the formation of the $O_2$ molecule, are shown in the lower panel. (Reprinted from ref. [65] with permission from Journal of American Chemical Society. Copyright © 2013, American Chemical Society).

Metal oxide plays a crucial role in catalytic OER reactions. Typical examples are molecular metal-oxo clusters, solid-state materials, and oxo-bridged metal dimers. After the first report by Nocera and co-workers [147] the cobalt phosphate (CoPi) catalyst gained much attention owing to its ease of synthesis, robustness, durability, self-healing, and high activity at neutral pH [145,146,150,156]. Hybrid quantum mechanics/molecular mechanics (QM/MM) based DFT calculation on the CoPi catalyst by Wang et al. [157] shows that two $Co^{IV}(O)$ metal oxo groups undergo direct coupling with the neighboring oxygen to form O–O bond. With cubane type complex and other related complexes it was shown that the formation of M(IV)–O$^{\bullet}$ state is the crucial step for OER [158–160].

Bajdich et al. [68] studied the bulk and surface structure of cobalt oxide catalysts using DFT+U to understand the electrocatalytic OER activity. From the calculated Pourbaix diagram (Figure 22) it was shown that the CoOOH phase is the most stable phase under typical OER conditions (pH = 12–13 and $U > 1.23$ to ~1.7 V vs. RHE). The energies of low-index surfaces of β-CoOOH: (0001), (01$\bar{1}$2), and (10$\bar{1}$4) (Figure 23) were calculated, which depend on pH and applied potential. One mL of $H_2O$ covered the (10$\bar{1}$4) surface below $U_t$ ($U_t = 1.90$ V $- 0.059$pH) while 1 mL of the $O_t$ (t denotes the top position) terminated (01$\bar{1}$2) surface is more stable above $U_t$. Theoretically, the calculated overpotential for the (10$\bar{1}$4) and (01$\bar{1}$2) surfaces are 0.48 V and 0.80 V, respectively. The representative OER mechanism for the (10$\bar{1}$4) surface is shown in Figure 24. At the OER, Co is in +3 oxidation state in the more active (10$\bar{1}$4) surface while it is in +4 state in the less active surfaces. The overpotential for OER is further decreased by doping with Ni to form $Ni_yCo_{1-y}O_x$ thin films.

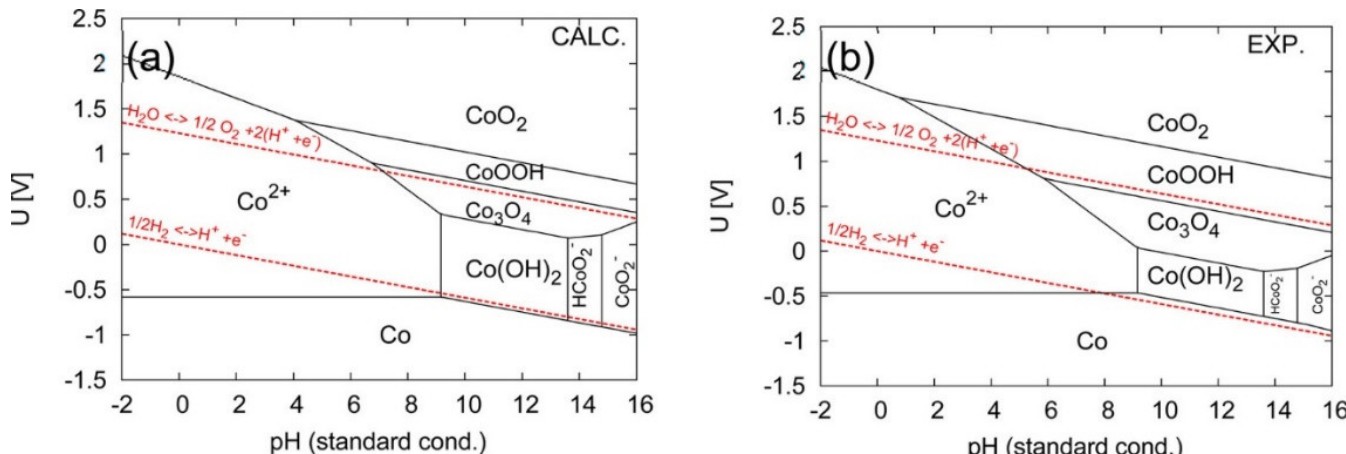

**Figure 22.** Pourbaix diagrams of bulk phases based on (**a**) the calculated formation free energies of solid compounds and the corrected experimental free energies of aqueous ionsand (**b**) based only on experimental formation free energies. (Reprinted from ref. [68] with permission from Journal of American Chemical Society. Copyright © 2013, American Chemical Society).

Chen et al. [161] studied the OER on the (110) surface of the spinel cobalt oxide, $Co_3O_4$, using the DFT+U method. The surface contains positively (A) and one negatively (B) charged terminations (Figure 25). Partial or fully hydroxylated $Co_3O_4$ (110) surface is formed through the dissociative adsorption of water molecules. The computational investigations were performed on the neutral species and hence solvent effects were marginalized. It was found that the deprotonation of adsorbed OH* to form O* is the more challenging step. The A terminated surface is more active compared to the B. The reason is that on the A-terminated surface the large density of cobalt was found near the Fermi level, which reduces the overpotential through the stabilization of O* species. The results are consistent with the general studies of OER on metal oxide surfaces [2].

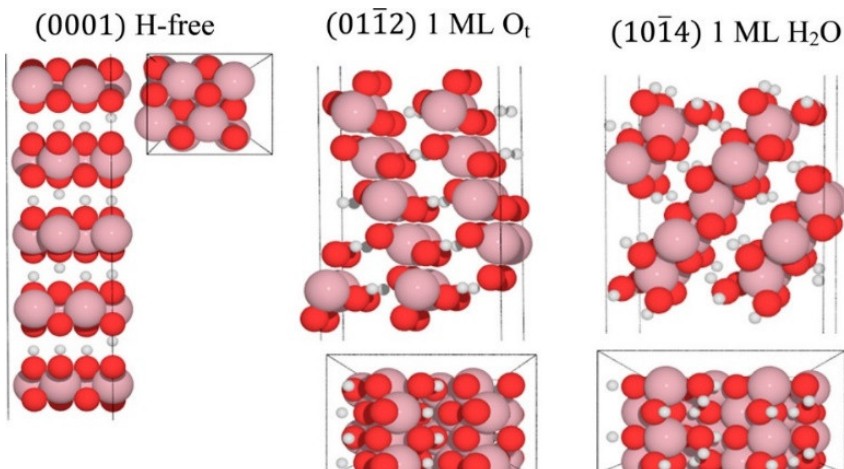

**Figure 23.** Side- and top-views of the optimized geometries for the lowest-energy surfaces of β-CoOOH represented as 5-layer symmetric slabs and used to determine the surface energies. The small white spheres represent H, the red spheres represent O, and the large pink spheres represent Co atoms. (Reprinted from ref. [68] with permission from Journal of American Chemical Society. Copyright © 2013, American Chemical Society).

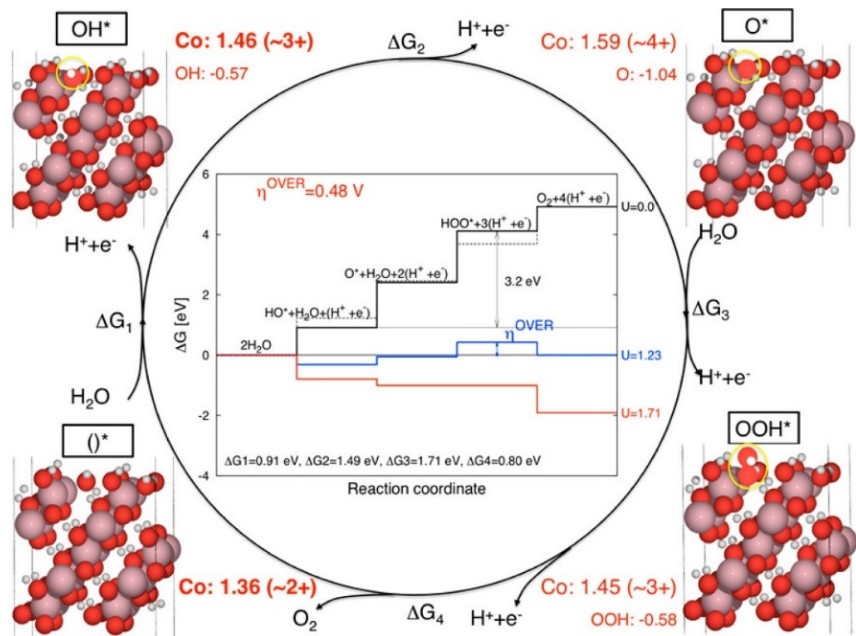

**Figure 24.** Schematic of the OER on the (10$\bar{1}$4) surface. The inset shows the free-energy landscape compared to an ideal catalyst (dashed-line) for zero pH. Reaction 3 is the potential-limiting step. For U > 1.71 V, all steps are thermodynamically accessible. (Reprinted from ref. [68] with permission from Journal of American Chemical Society. Copyright © 2013, American Chemical Society).

In a different proposed mechanism it was shown that after the electrochemical formation of $Co^{IV}$=O it can undergo non-electrochemical proton/electron hopping (Figure 26) to form an oxo pair $Co^{IV}$(=O)-O-$Co^{IV}$=O [162]. Following the formation of $Co^{IV}$=O, a water nucleophilic attack generates the hydroperoxide $Co^{III}$–OOH. Ultimately $O_2$ is evolved through the formation of $Co^{III}$–OO upon an encounter with another oxo group. The non-electrochemical reactions were found to proceed with a low kinetic barrier and were thermodynamically downhill.

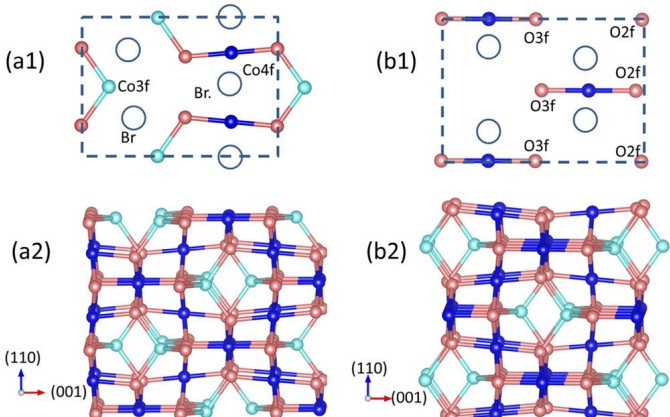

**Figure 25.** Stick-and-ball model of the $Co_3O_4$ (110) surface structure. Top (**a1,b1**) and side (**a2,b2**) views of A (**left**) and B (**right**) terminations. In the top views, empty blue circles indicate water and hydroxyl group adsorption sites. Dark and pale-blue spheres indicate Co cations, which in the bulk are octahedrally and tetrahedrally coordinated, respectively. Red spheres indicate oxygen anions. (Reprinted from ref. [161] with permission from Journal of Physical Chemistry Letters. Copyright © 2012, American Chemical Society).

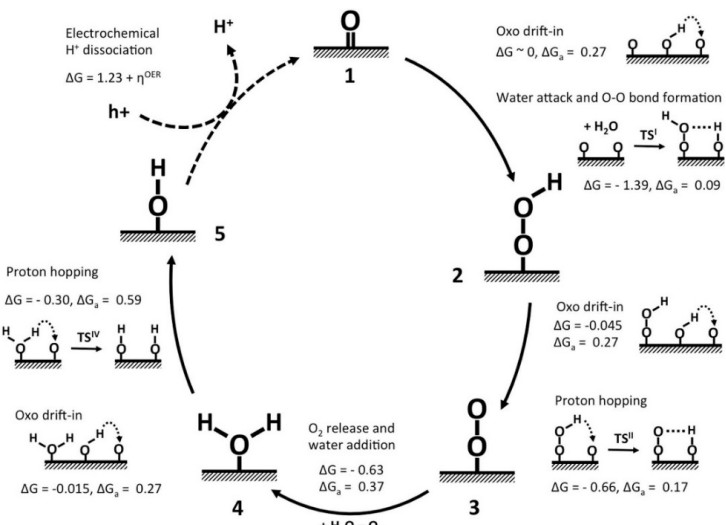

**Figure 26.** Proposed mechanism of oxidation catalysis by the water attack and proton hopping pathway. The Gibbs free energy reduction ($\Delta G$) and activation barriers ($\Delta G_a$) of the evolution are given in eV. For simplicity, only one oxo site (where molecular oxygen evolves) is demonstrated in the big circle (intermediates **1–5**). However, all calculations for water attack and proton jumping are performed with the assumptions that at least one adjacent oxo is available. In this scheme, only the $Co^{III}$–OH to $Co^{IV}$=O would be of electrochemical nature (requiring the transfer of an external hole). Other conversions (water addition and surface proton/electron hopping) are spontaneous (not electrochemical). (Reprinted from ref. [162] with permission from Journal of physical ACS Catalysis © 2014, American Chemical Society).

The isotope effect on the mechanism of OER on cobalt oxyhydroxide (CoOOH) was studied [163]. The OER mechanism does not change upon changing the KOH solution to KOD as the Tafel slope remains fixed at 60 mV dec$^{-1}$. The experimental potential to achieve 1 mA cm$^{-2}$ (vs RHE or RDE) increased by 30 mV in the KOD solution compared to KOH. This was attributed to the equilibrium isotope effect (EIE), which affects the oxidation potential of Co(III) to Co(IV) couple (i.e., $CoOOH \rightarrow CoO_2$). The DFT calculated shift for the EIE is 33.3 mV.

The OER was performed on crystalline spinel-type $Li_2Co_2O_4$ ($Co^{3+}$) material. During CV cycles, spontaneous de-lithiation increased the OER activity. [164] DFT calculations suggest that de-lithiation increased the oxygen vacancy formation by decreasing the energy needed to create the vacancy. The reconstruction pathway includes: (i) formation of $Co^{4+}$ and oxidized oxygen ions due to delithiation. (ii) the oxidized lattice oxygen leaves the lattice as $O_2$ and creates oxygen vacancies. (iii) These vacancies are filled by hydroxydes from the solution. Throughout the process $Co^{IV}$ plays a crucial role in activating the oxidized oxygen.

Peng et al. [165] performed DFT+$U$ for the OER at $Co_3O_4$(001) surface. The *Pourbaix* diagram was constructed as a function of oxygen partial pressure as well as pH and applied voltage to understand the stability of different surface terminations. The B-layer containing octahedral Co and O terminations shows the lowest overpotential of 0.46 V for OER (Figure 27). The active site of Co possesses a high oxidation state of +4 and p-type conducting character. The overpotential decreases to 0.34 V from 0.46 V upon doping with Ni. Ni doping takes place at the octahedral site of the B-layer whereas Fe doping occurs at the tetrahedral site of the A-layer. Fe doping changes the overpotential from 0.46 to 0.37 V. The crucial factors on which the overpotential depends are the transformation of *OH (B-layer) to *OOH (A-layer). Explicit solvent considerations increase the potential slightly (0.2 V for B-layer and 0.4 V for A-layer). The scaling relationship, which is the difference in free energies of formation of *OOH and *OH (i.e., $\Delta G_{*OOH}{}^b - \Delta G_{*OH}{}^b$), is also satisfactory. In the volcano type plot, i.e., the plot of overpotential vs. $\Delta G_{*O}{}^b - \Delta G_{*OH}{}^b$ the doped systems is close to the top (Figure 28).

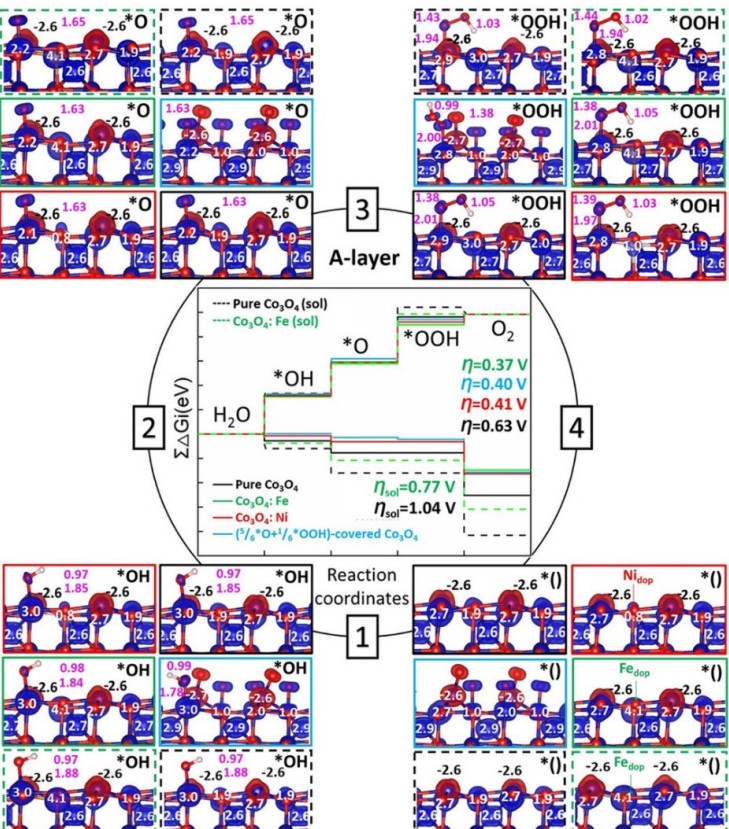

**Figure 27.** OER cumulative free energies $\sum \Delta G_i$ of step(1)–(4) at the A-layer. The side view of the intermediates with the corresponding spin-density is displayed. Additionally, the magnetic moments are given in $\mu_B$ in black or white and bond lengths in Å in magenta. Black, green, and red frames denote the undoped, Fe-, and Ni-doped $Co_3O_4$(001) surfaces, respectively; the dashed lines indicate the calculations including solvation effects. (Reprinted from ref. [165] with permission from the ACS Catalysis © 2021, American Chemical Society).

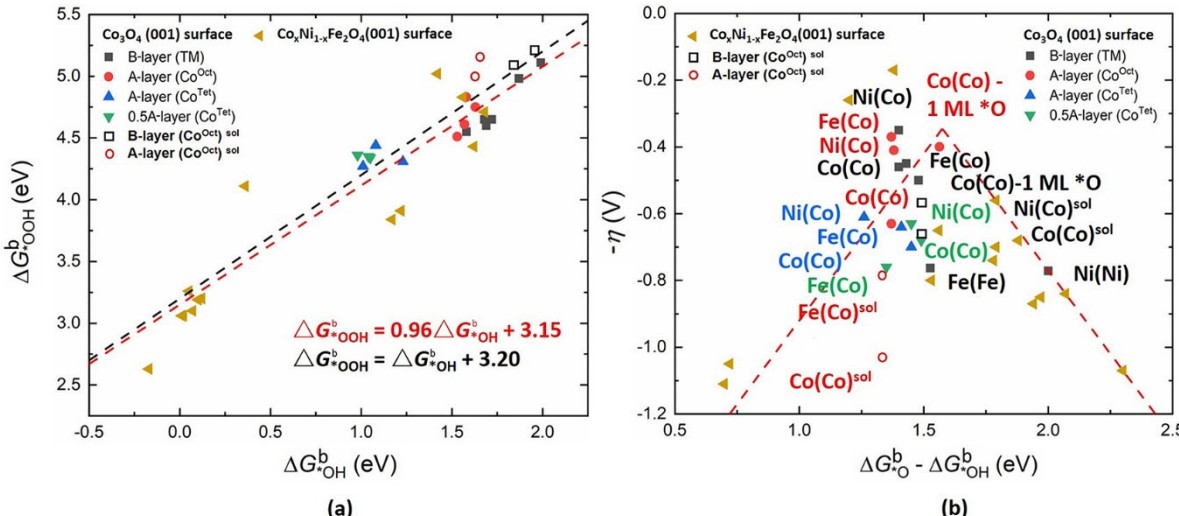

**Figure 28.** (**a**) Scaling relationship between $\Delta G_{*OOH}^{b}$ and $\Delta G_{*OH}^{b}$ for different reaction sites at the B-, A-, and 0.5A-terminations of the $Co_3O_4$(001) surface. Red and black dashed lines indicate the result of the linear fit and unity slope, respectively; (**b**) Volcano plot of the negative overpotential as a function of ($\Delta G_{*O}^{b} - \Delta G_{*OH}^{b}$). The red dashed line is calculated according to equation ($\eta^{\delta C} = [\max((\Delta G_{*O}^{b} - \Delta G_{*OH}^{b}), 3.15\,\text{eV} - (\Delta G_{*O}^{b} - \Delta G_{*OH}^{b}))/e] - 1.23\,\text{V}$). The results of $Co_xNi_{1-x}Fe_2O_4$(001) [166] is represented by the yellow triangles using of the RPBE functional. (Reprinted from ref. [165] with permission from Journal of physical ACS Catalysis © 2021, American Chemical Society).

$Co_3O_4$ is a normal spinel (space group $Fd\bar{3}m$) [167] where high spin $Co^{2+}$ occupy tetrahedral holes and low spin $Co^{3+}$ occupy octahedral holes. The advantages of $Co_3O_4$ for catalytic applications are its availability, stability [168], and p-type semiconducting property (low conductivity) [169]. In acidic medium, the OER potential of $Co_3O_4$ is only 0.20–0.25 V higher than $RuO_2$ [170]. There are opposing and mixed statements regarding the OER mechanism of $Co_3O_4$: (i) Menezes et al. [171] stated that $Co^{oct}$ is the active site, (ii) Wang et al. [172] suggested $Co^{tet}$, and, (iii) according to Plaisance and van Santen [173], oxygen bridge adjacent Co sites are the only active site. The incorporation of bimetallic systems of the type $M_xCo_{3-x}O_4$ (M=Li, Fe [154,174], Ni [175] and Cu [176]) were investigated with special emphasis on Fe and Ni [177–180].

Application of the DFT+$U$ method accurately predicts the electronic properties. In this regard it was shown that use of the GGA function (PBE0) gives a band gap of 1.96 eV (experimental value is 1.6 eV) while using only the hybrid functional PBE0 calculates the value to 3.42 eV [180]. PBE0+$U$ shows that the (1 0 0) surface terminations are more stable than the (1 1 0) and (1 1 1) terminations. The B-layer with octahedral Co and A-layer with tetrahedral Co are the most stable [180,181]. The 0.5A-layer was also investigated. The equilibrium structure contains (100) and (111) [182]. Interestingly, nanorods formation with a (110) surface was found to be of major interest owing to its water adsorption properties [161]. One important aspect of this catalyst is that it undergoes surface reconstruction during OER [37,183–185]. He et al. [186] demonstrated that the surface active electron density band center of $Co_3O_4$ can be upshifted by the application of an argon-ion irradiation method. The adsorption capability of the oxo group is significantly enhanced by such an upshift.

Selcuk et al. [187] studied the effect of Coulomb U term on the stabilities of the A and B surfaces in the presence of oxygen gas and water vapor. At all potentials (U = 0, 3.0, and 5.9 eV) under ambient conditions and low temperature the hydrated B surface with exposed octahedral Co site is more stable. However, apart from ambient conditions, low U stabilizes the B surface while high U stabilizes the A surface. U = 3.0 eV best describes the electronic structure and surface reactivity. The doped CoOOH-based catalyst (FeCoW oxyhydroxides) shows promising OER activity with an overpotential of

0.19 V [188,189]. To understand the enhanced OER activity of CoOOH, an extra step is introduced in the reaction mechanism [190]. The deprotonation of adsorbed water is separated into two steps: (a) $H_2O^* \leftrightarrow (HO + H)^*$ and (b) $(HO + H)^* \leftrightarrow HO^* + H^+ + e^-$, which reduces the kinetic barrier of the reaction. The DFT-based OER activity was demonstrated on a pristine CoOOH $(01\bar{1}2)$ surface. Owing to the inert nature of the surface, the oxygen intermediates are weakly adsorbed on the surface, which leads to a high overpotential of 0.65 V where the deprotonation of water to form OH* is the rate determining step. CoOOH (Vo-CoOOH) with slightly modified oxygen vacancies improves the adsorption of the intermediates OH* and O*. The strengthening of the bond is justified from the initial bond lengths of pristine Co-O and Cov-O (where Cov = oxygen vacancies adjacent Co), which are 1.64 and 1.61 Å, respectively. Further the deprotonation of water takes place in two steps as mentioned above and the formation of OOH* from O* becomes rate determining (Figure 29). The calculated overpotential becomes 0.262 V at 10 mA cm$^{-2}$.

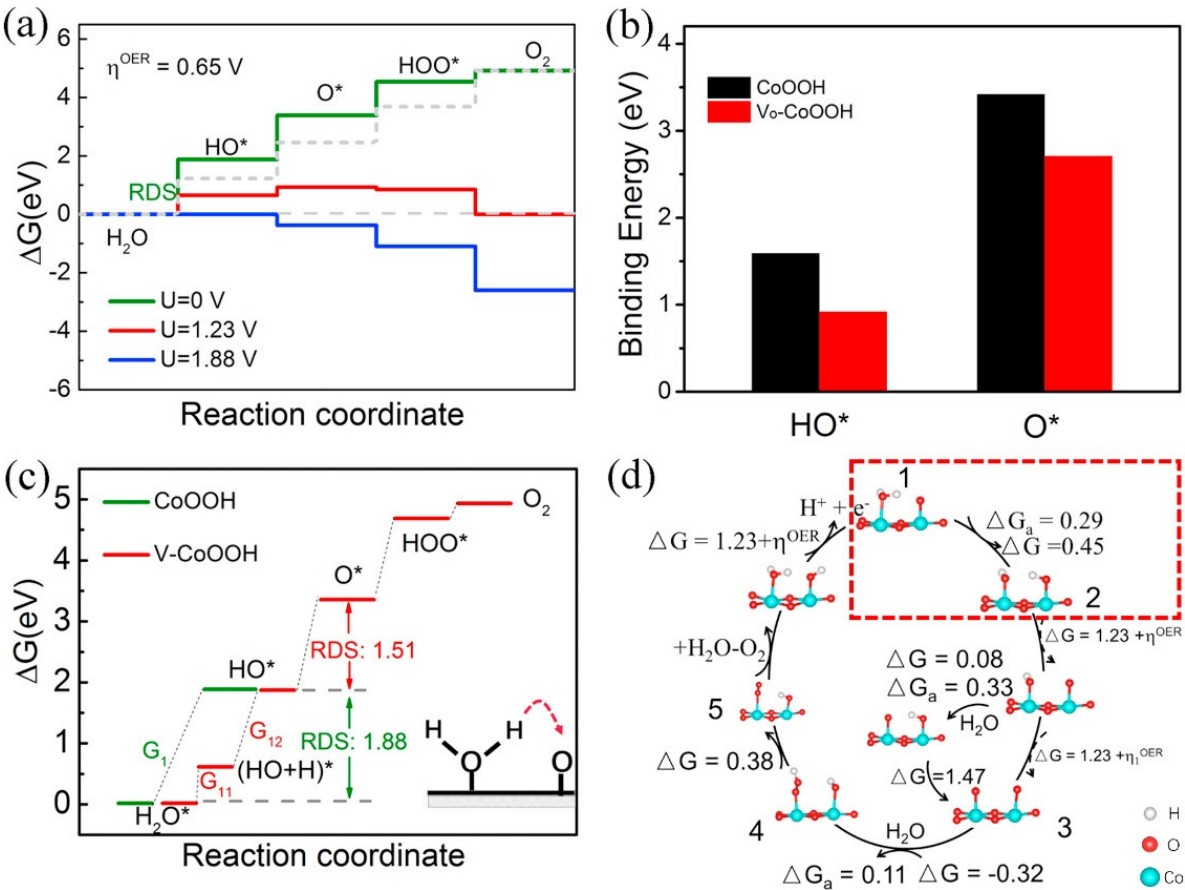

**Figure 29.** (**a**) Free energy diagram for the 4-step OER path under different positive potential U on pristine; (**b**) calculated binding energy for pristine and vacancy modified CoOOH; (**c**) Free energy diagram for 5-step OER path on oxygen vacancy modified CoOOH $(01\bar{1}2)$ surface; and (**d**) Schematic illustration of the OER pathways, the Gibbs free energy reduction ($\Delta G$) and activation barriers ($\Delta G_a$) of the evolution are given in eV. Cyan spheres-cobalt, red-oxygen, and white-hydrogen. (Reprinted from ref. [190] with permission from Nano Energy © 2017, Elsevier).

Further reactivity was assessed from the DOS analysis (Figure 30). By increasing the energy of the d-state ($E_d$) relative to the Fermi level ($E_f$), the filling of the antibonding state decreases and hence the adsorption is stronger. Vo-CoOOH possesses impurities due to the presence of dangling bonds (Figure 30) in the surface Co octahedron, which increase the energy of $E_d$ and hence antibonding filling decreases, ultimately leading to increased bonding.

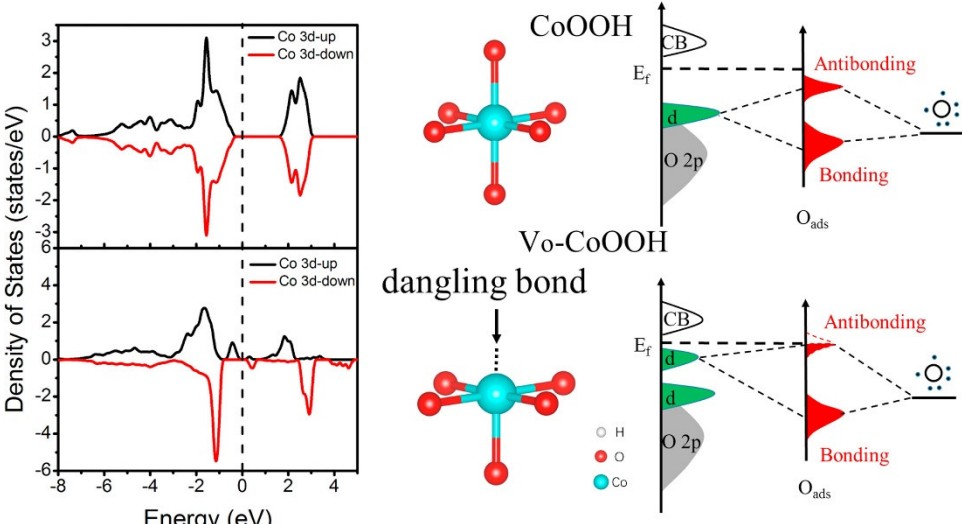

**Figure 30.** Calculated density of states (DOS) and corresponding schematic bond formation of atomic oxygen through coupling of O 2p to the highest occupied d-state on pristine and oxygen vacancy modified CoOOH. Cyan spheres—cobalt, red—oxygen. (Reprinted from ref. [190] with permission from Nano Energy © 2017, Elsevier).

So far, a four-step mechanism is frequently studied in heterogeneous OER mechanism. Recently, Curutchet et al. [191] proposed three competitive mechanisms of OER on a -CoOOH (10–14) hydrated surface involving one OH-site and two OH-sites: (i) Mechanism I: A typical four-step process with an additional non-electrochemical step, (ii) Mechanism II: Involves two sites of water oxidation, and (iii) Mechanism III: a two sites process but involving tetraoxidane intermediates. These are schematically shown in Figure 31. Mechanism II has the lowest overpotential with no activation barrier and hence is the most preferred. Mechanism IV with the tetraoxidane intermediates requires the highest activation barrier to be overcome and hence is the least feasible.

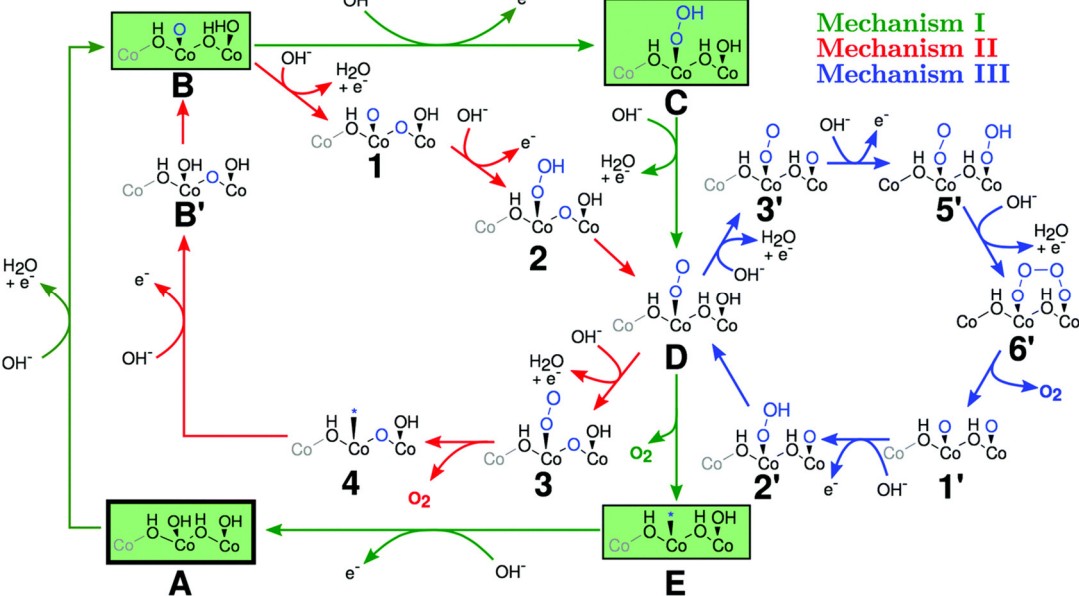

**Figure 31.** Three mechanisms in competition for water oxidation. The four intermediates frequently reported in the literature are highlighted in green. (Reprinted from ref. [191] with permission from Physical Chemistry Chemical Physics © 2020, Royal Society of Chemistry).

The electronic configuration of the transition metal in a catalyst plays a crucial role in the mechanism [192–195]. Involvement of $e_g$ orbitals increases the activity. Ligand field theory is used to understand the electronic structure of CoOOH. Due to the higher crystal field splitting than the pairing energy, the Co$^{III}$ in CoOOH remains in the low spin configuration ($t^6_{2g}e^0_g$) [196]. The fully occupied $t_{2g}$ orbitals lead to large energy for the rate limiting formation of adsorbed *OOH species due to smaller conductivity [197]. Huang et al. [197] demonstrated that due to structural deformities the electron configuration of Co$^{III}$ in ultrathin CoOOH nanosheets changes from $t^6_{2g}e^0_g$ to $t^5_{2g}e^{1.2}_g$ which ultimately gives enhanced OER activity. Li et al. [198] have shown through DFT calculations that strain engineering can be utilized to enhance the OER activity by two-dimensional (2D) CoOOH. It was shown (Figure 32) that increasing the tension by 9% the electronic configuration of Co$^{III}$ changes from low spin (LS: $t^6_{2g}e^0_g$) to high spin (HS: $t^4_{2g}e^2_g$). LS CoOOH is a poor catalyst, as it requires large energy (1.35 eV) for the release of O$_2$. On the other hand, LS CoOOH requires only 0.03 eV energy for the O$_2$ releasing step and hence acts as a good catalyst. The overpotential calculated for the LS CoOOH is 0.66 V via the hydroxide ion attack mechanism while the intramolecular coupling does not occur. In HS CoOOH the overpotential values are 0.32 and 0.50 V for the hydroxide ion attack mechanism and the intramolecular coupling mechanism, respectively, which is comparable to the best-known catalysts (0.25 to 0.4 V).

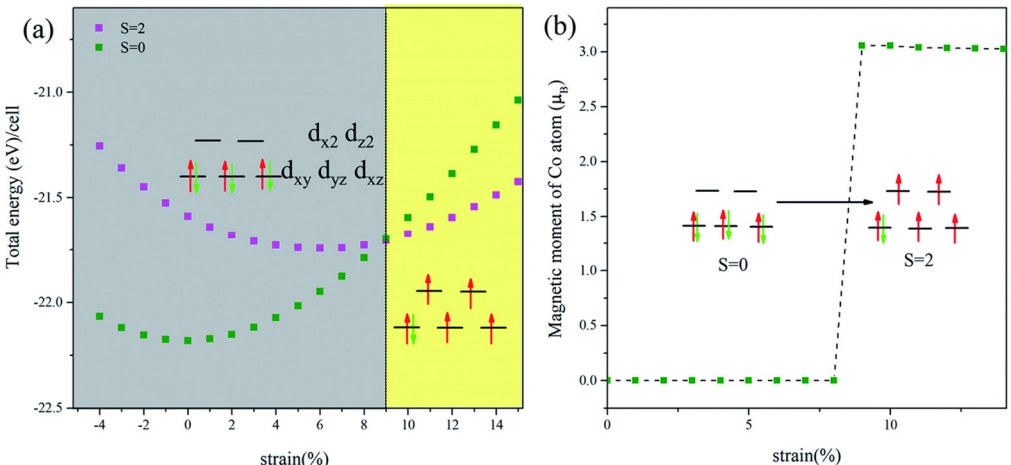

**Figure 32.** (**a**) The total energy of CoOOH versus strain and (**b**) magnetic moment of Co$^{3+}$ ions versus strain. (Reprinted from ref. [198] with permission from Journal of Materials Chemistry A © 2021, Royal Society of Chemistry).

The OER mechanism was studied for the Co$_3$O$_4$ spinel for the thermodynamically most stable (001) surface considering single-Co and dual-Co sites. Two more surfaces (110) and (311) were studied by Plaisance et al. [173] and were found to be more active than the (001) surface. The single-Co and dual-Co sites were found to be active in (110) and (311) surfaces, respectively.

García-Mota et al. [170] studied the OER mechanism on Co$_3$O$_4$(001) and β-CoOOH (01$\bar{1}$2) surfaces. They have shown the importance of Hubbard-$U$ correction to DFT (i.e., DFT+$U$) to correlate theoretical results with the experimental results. Pourbaix diagrams were constructed for a range of potentials and pH values. In an acidic environment, bulk dissociation of cobalt takes place spontaneously. Thus, alkaline conditions of the experiments were implemented to exclude the dissolution. Up to 1.8 $V_{RHE}$ ($V_{RHE}$, the voltage, is reported vs. the reversible hydrogen electrode), the Co$_3$O$_4$ surface is stable, and, above this, the potential formation of OOH occurs on the oxygen-covered Co$_3$O$_4$ surface. The trend remains the same with the CoOOH surface. The binding of O* to the surface becomes 1.1 eV weaker upon application of DFT+$U$. However, this value for OH* and OOH* binding energies is 0.6 eV. Thus, it was suggested that formation of OH* from

water is the rate determining step, as the associated change is high in this case. The next elementary step is the OH* to O* transformation. Therefore, in the OER volcano plot, the position shifts from the strong binding leg to the weak binding leg in $Co_3O_4$ and $\beta$-CoOOH (Figure 33). The calculated overpotential for $Co_3O_4$ and $\beta$-CoOOH were found to be similar. This is attributed to the similar local coordination environment of Co in both materials. The Hubbard-$U$ correction increases the overpotential from 0.41 to 0.76 V and makes it consistent with respect to the $RuO_2$ system. The Bader change on the Co active sites of CoOOH and $Co_3O_4$ are calculated to be ~+1.6, and +1.44, respectively.

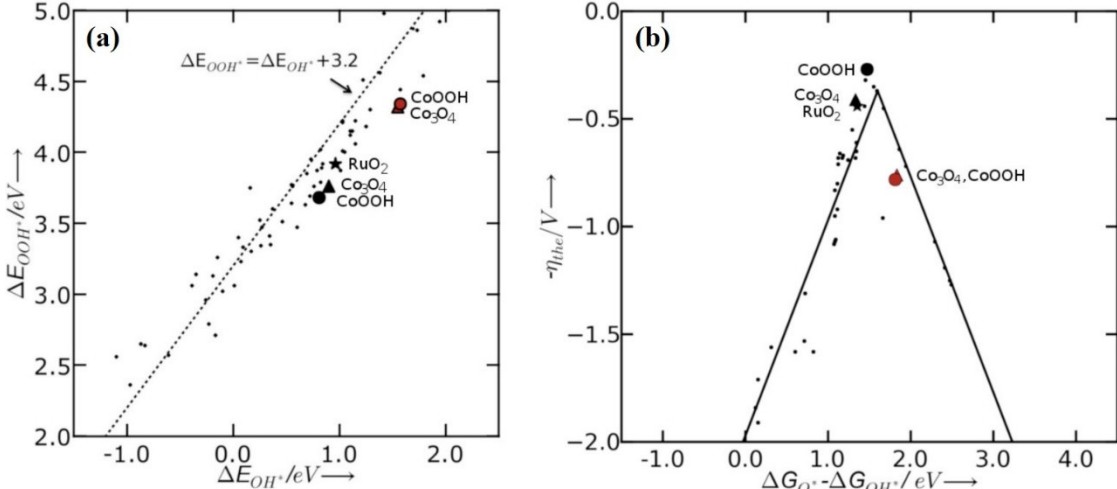

**Figure 33.** (a) Adsorption energies of OOH* ($\Delta E_{OOH*}$) as a function of the adsorption energies of OH* ($\Delta E_{OH*}$) on the oxygen-covered $Co_3O_4$(001) and CoOOH (01$\bar{1}$2) surfaces, shown in triangles and circles, respectively. The black and red colors correspond to the calculations performed with RPBE and with RPBE+$U$, respectively. The small black circles correspond to $\Delta E_{OOH*}$ as a function of $\Delta E_{OH*}$ on the oxide surfaces. The black star indicates the position of $RuO_2$(110). The binding energies of OOH* and OH* scale according to the relation, $\Delta E_{OOH*} = \Delta E_{OH*} + 3.2$, with 95% of the points being within $\pm 0.4$ eV. (b) Calculated theoretical overpotentials ($\eta_{the}$) plotted as a function of ($\Delta G_{O*} - \Delta G_{OH*}$) for oxygen-covered $Co_xO_y$ surfaces. The small black circles correspond to $\eta_{the}$ as a function of ($\Delta G_{O*} - \Delta G_{OH*}$) on rutiles and perovskites. The volcano curve is established by using the scaling relation between ($\Delta G_{OOH*} - \Delta G_{O*}$) and ($\Delta G_{O*} - \Delta G_{OH*}$). (Reprinted from ref. [170] with permission from The Journal of Physical Chemistry C © 2012, American Chemical Society).

A comparative stability study was performed by Chen et al. [199] for crystalline cobalt oxides and hydroxides: CoO, $Co(OH)_2$, $Co_3O_4$, CoO(OH), and $CoO_2$ (Figure 34) in an electrochemical environment using DFT with on-site Coulomb repulsion. The following reactions were considered to calculate the changes in free energies:

$$CoO(s) + H_2O(l) \rightarrow Co(OH)_2(s) \tag{42}$$

$$Co(OH)_2(s) \rightarrow CoO(OH)(s) + H^+ + e^- \tag{43}$$

$$Co_3O_4(s) + 2H_2O(l) \rightarrow 3CoO(OH)(s) + H^+ + e^- \tag{44}$$

$$CoO(OH)(s) \rightarrow CoO_2(s) + H^+ + e^- \tag{45}$$

In low pH reducing conditions, ($V < 0$ vs. SHE) $Co(OH)_2$ is thermodynamically stable, while at high pH and potential, ($V < 1.23$ eV vs. SHE) CoO(OH) and $CoO_2$ are stable. They showed a corroborating experimental result on the (0001) natural cleavage surface of CoO(OH) in the presence of the $CoO_2{}^{x-}$ (x = 0–0.5) layer under oxidizing conditions. In the OER mechanism, the layered oxide transformation of adsorbed water to OH* is energetically the most demanding and leads to high overpotential. Thus, layered $CoO_2{}^{x-}$ is not active for the OER. In comparison with active $Co_3O_4$, the O* → OOH* is the rate

determining step with a smaller barrier. The high activity of the cobalt oxide electrode toward OER was attributed to the incomplete transformation of the catalytically active $Co_3O_4$ phase, having possibly more surface defects on converted $CoO_2^{x-}$ layers.

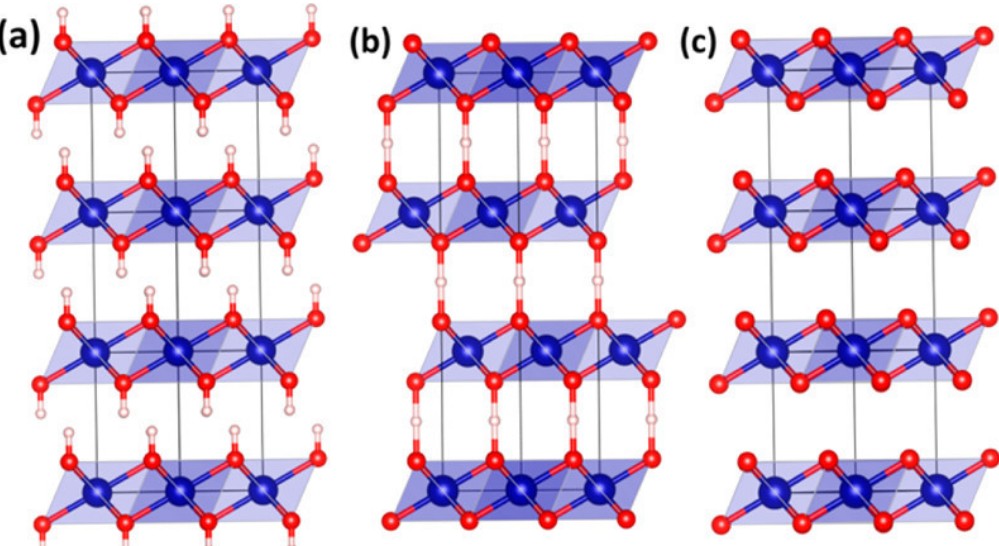

**Figure 34.** Layer structure of (**a**) $Co(OH)_2$, (**b**) $CoO(OH)$, and (**c**) $CoO_2$. The main difference among these structures is the number of protons between the $CoO_2$ layers. (Reprinted from ref. [199] with permission from The Journal of Physical Chemistry C © 2013, American Chemical Society).

The effect of cation substitution and surface termination on the $Co_xNi_{1-x}Fe_2O_4(001)$ surface ($x$ = 0.0, 0.5, 1.0) was studied for the OER reaction. Fe, Co, Ni, and oxygen vacancies were considered as the active site on three terminations: (i) octahedral Co at the B-layer, (ii) octahedral Ni at the B-layer, and (iii) half and full monolayer of Fe (0.5A and A-layer, respectively). An equal concentration of Ni and Co, i.e., $Co_{0.5}Ni_{0.5}Fe_2O_4(001)$ provides the lowest theoretically calculated overpotential of 0.26 V. Analysis of the electronic properties show that the presence of an additional Fe layer stabilizes the baulk-like oxidation state of +2 for Co and Ni at the A-layer. At the B-layer, the +3 oxidation states are stabilized. An H-bond formation of intermediate OOH with neighboring O was observed due to the unusual relaxation pattern, which lowers the reaction free energy and deviates the scaling relationship. This is an excellent example of surface tuning to lower the overpotential and improve the OER.

The $Fe^{3+}$ doping on the $Co_3O_4$ spinel forms mixed spined $Co_2FeO_4$ with the composition $(Co_{0.72}Fe_{0.28})_{Td}(Co_{1.28}Fe_{0.72})_{Oct}O_4$, which was grown as nanoparticles over N-doped carbon nanotubes (NCNTs) [174]. This catalyst shows superior activity compared to the noble metal catalysts. Introduction of Fe to the framework of $Co_3O_4$ delocalize the Co 3d electrons and changes the spin state.

### 3.4. Nickel Oxide-Based Catalyst

In the last decade scientists have identified pure and doped $NiO_x$ as catalysts of the OER [200,201]. Their utilization in alkaline batteries was long known [202,203]. First, a theoretical study on nickel hydroxide was performed by Hermet et al. [204] to understand the magnetic, electronic, dielectric, dynamic, and elastic properties. During charging, β-NiOOH and overcharging leads to the formation of γ-NiOOH, as obtained from the Bode's diagram (Scheme 1). It is generally believed that β-NiOOH is the active phase for OER [205]. However, Bediako et al. [206] pointed out that γ-NiOOH might be more active.

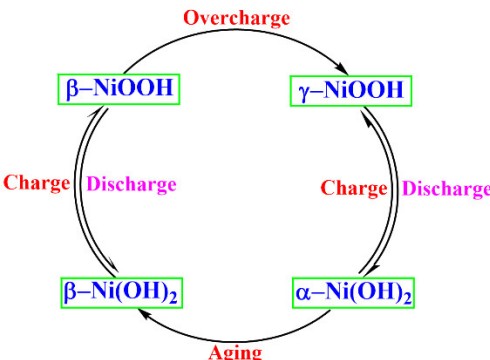

**Scheme 1.** Bode's Diagram for Ni(OH)$_2$–NiOOH redox transformations.

Li and Selloni investigated β-NiOOH(01$\bar{1}$5) and γ-NiOOH(101), Fe-doped β-NiOOH(01$\bar{1}$5) and γ-NiOOH(101), NiFe$_2$O$_4$(001), and Fe$_3$O$_4$(001) surfaces as electrocatalysts for OER [206]. Fe-doped β-NiOOH(01$\bar{1}$5) was found to be the most active with an overpotential (η = 0.26 V) lower than that of RuO$_2$ (0.36 V) [50,63], followed by NiFe$_2$O$_4$(001) (η = 0.42 V). A recursive trial-and-error approach was utilized to find the most likely pathways by adding H$_2$O to the surface and then removing H (proton/electron pair) stepwise until an O$_2$ species was produced. The trend of the overpotential follows the order: Fe-doped β-NiOOH (0.26 V) < NiFe$_2$O$_4$ (0.42 V) < β-NiOOH (0.46 V) < Fe-doped γ-NiOOH (0.48 V) < γ-NiOOH (0.52 V) < Fe$_3$O$_4$ (0.70 V). The small difference in the calculated overpotential (0.06 V) of β-NiOOH and γ-NiOOH answers the question as to why both are considered active phases for the OER.

For β-NiOOH(01$\bar{1}$5) the deprotonation of adsorbed water is the rate determining step that leads to an overpotential of 0.46 V. Then, a second proton is released, followed by the O-O bond formation between O adatom and a surface lattice O$_{3c}$ (where O$_{3c}$ is a three coordinated oxygen). Adjacent to the formed O-O bond, a water molecule gets adsorbed to an exposed Ni$_{5c}$ (where Ni$_{5c}$ is five coordinated nickel). After the fourth deprotonation O$_2$ is formed and desorbed, and the water molecule gets adsorbed subsequently on the exposed Ni$_{5c}$. The OER mechanism on the Fe-doped β-NiOOH(01$\bar{1}$5) is similar. However, the second proton release is the RDS with η = 0.26 V. Meanwhile, in pure γ-NiOOH(101), the RDS is the third deprotonation.

Fe-doped γ-NiOOH(101) shows a slightly different mechanism. Here, two adjacent water molecules lose two protons subsequently. The second deprotonation is the RDS. Then, the third proton is released from an adsorbed OH and subsequently forms an O-O bond with adjacent surface lattice O$_{3c}$. The other steps remain the same. Fe$_3$O$_4$(001) shows a similar mechanism with a high overpotential η = 0.70 V.

It was believed that upon removal of a proton from β-Ni(OH)$_2$ the β-NiOOH is obtained [50]. Delahaye-Vidal et al. [50] revealed the production mosaic textures during the first oxidation cycle from β-Ni(OH)$_2$ to β-NiOOH due to an irreversible microstructural transformation, which was also confirmed by HRTEM [50]. Doubling of the *c* axis of β-NiOOH was identified by Casas-Cabanas et al. [50]. They concluded that NiO$_2$H sheets with "ABCA" stacking are formed. However these experimental X-ray results are not in agreement with the DFT results [207,208]. NiO$_2$ sheets stacked in an "ABBCCA" pattern to form layered γ-NiOOH are suggested by DFT [209]. DFT+U calculations at the PBE level show that the staggered structure is most stable with an U$_{eff}$ = 5.5 eV [209].

The global potential energy surface of β-NiOOH considering 12 Ni atoms, i.e., Ni$_{12}$O$_{24}$H$_{(12 \times x)}$ (*x* = 0–2) at a series of stoichiometries have been explored by SSW (Figure 35) [210]. The free energies of different phases as a function of electrochemical potential are studied using DFT (PBE+U, $U_{eff}$ = 5.5 eV). A more stable β-NiOOH phase with energy lowered by 0.02 eV/f.u was identified as alternate NiO$_2$ and Ni(OH)$_2$ local

layer structures. For other structures with composition $NiO_2H_x$ ($x > 0.5$ and $x < 0.5$), a layered structure was maintained, where perxo O-O bonding was featured.

$$Ni(OH)_2 \rightarrow NiOOH_x + (2 - x)(H^+ + e^-) \tag{46}$$

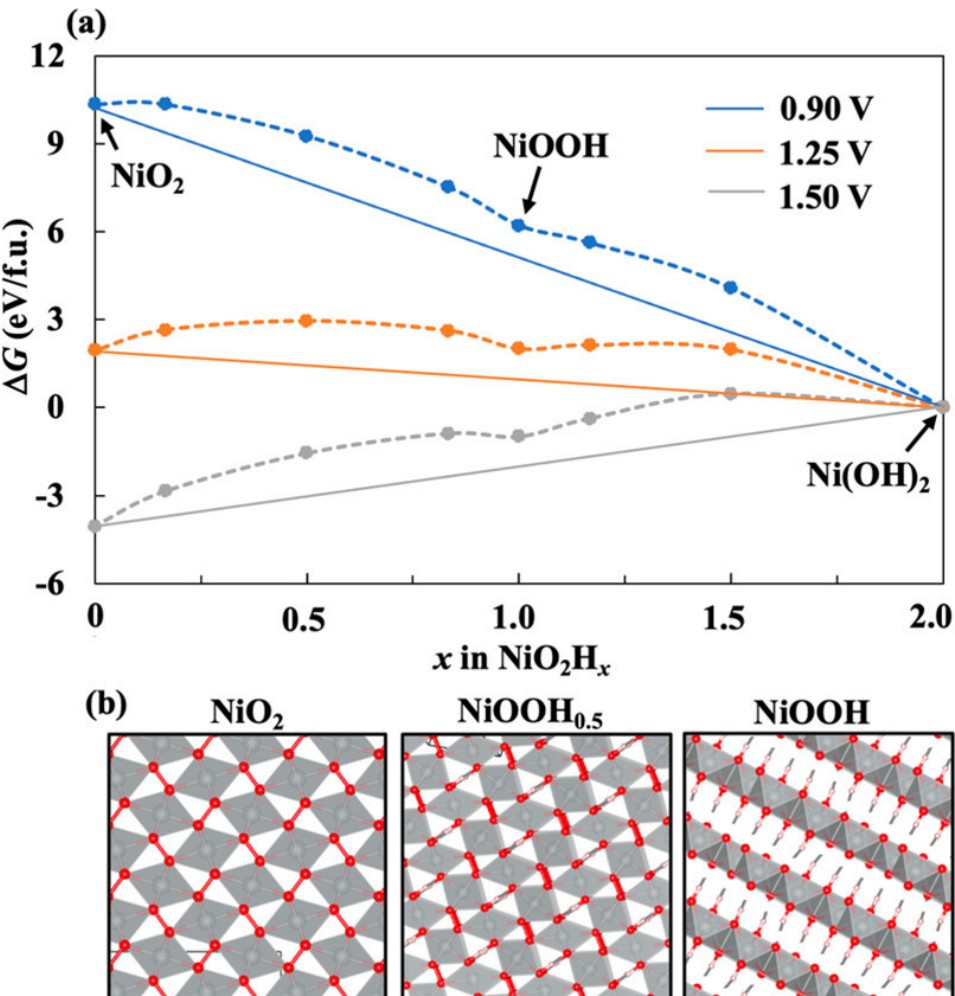

**Figure 35.** (**a**) Convex hull diagram of $NiO_2H_x$ at $U = 0.90$, 1.25, and 1.50 V vs. RHE. (**b**) Global minima of $NiO_2$, $NiOOH_{0.5}$, and $NiOOH$. Red balls—O; gray balls—Ni; white balls—H. The dashed lines in (**b**) represent the hydrogen bonds. (Reprinted from ref. [210] with permission from the Journal of Journal of Physical Chemistry C. Copyright © 2021, American Chemical Society).

Equation (46) was used to calculate the convex hull diagram of NiOOH phases. β-NiOOH is not a convex point in the diagram and hence tends to decompose into β-$Ni(OH)_2$ and tunneled $NiO_2$ above 1.25 V vs. SHE.

NiOOH behaves as a semiconductor with a band gap of 1.7–1.8 eV. Li and Selloni [207] reported on the results of the density of states (DOS) (Figure 36) to get the electronic structure details of alternate β-NiOOH using PBE+U, PBE0, and HSE06 functionals. It was stated from the PBE+U calculations that with no band gap in the spin-up component (the majority), a half-metallic with a ferromagnetic (FM) configuration is prevalent. The DFT+U results were unable to correlate the theoretical result correctly with the experimental one. However, a band gap of 1.5 and 2.5 eV is calculated at the hybrid HSE06 and PBE0 functionals. By implementing $G_0W_0$ calculations, Carter et al. reported a band gap of 1.96 eV staggered-II β-NiOOH, which corroborates well with the PBE/HSE06 results.

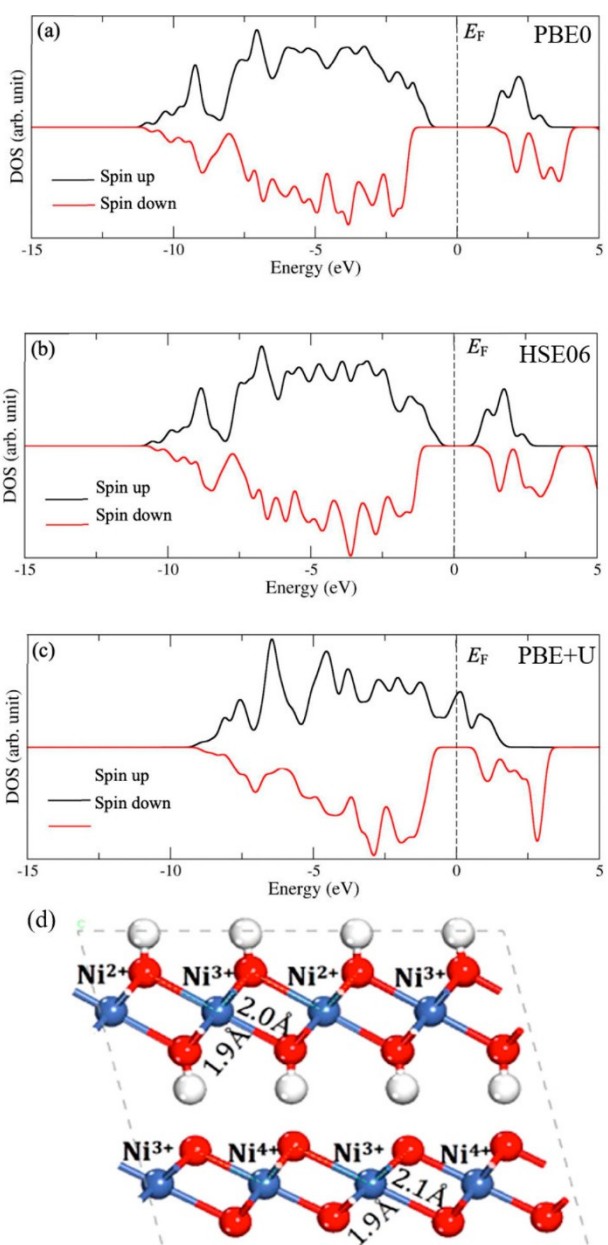

**Figure 36.** Electronic structures of alternate β-NiOOH. (**a**–**c**) The density of states with PBE0, HSE06, and PBE+U with $U_{eff}$ = 5.5 eV. The dotted lines represent the Fermi level. (**d**) The OS and bond lengths of Ni ions. (Reprinted from ref. [210] with permission from the Journal of Journal of Physical Chemistry C. Copyright © 2021, American Chemical Society).

The oxidation state of Ni is also crucial for their activity. According to the crystal field theory with an octahedral coordination environment, the electronic configuration $Ni^{(2+n)+}$ is $t_{2g}^6 e_g^{2-n}$. One can get an idea of the oxidation state from the magnetic moment and hence the spin polarization. $Ni^{II}$ and $Ni^{III}$ are paramagnetic with a net spin-polarization, but $Ni^{IV}$ is diamagnetic and has no spin polarization [207,211]. $Ni^{III}$ undergoes self-disproportionation to form $Ni^{II}/Ni^{IV}$. Toroker et al. [212] have shown that the PBE+U functional is sufficient for the geometry and electronic structure through a benchmark study using PBE, PBE+U, DFT-vdW (optPBE, optB86, and optB88), hybrid functionals (HSE06 and PBE0), and DFT-D2 (HSE06-D2 and PBE0-D2) functionals. Thus, the electronic structure of NiOOH is sensitive to functionals, and the use of hybrid functions better describe the properties.

The stabilities of various surfaces of β-NiOOH were calculated using the continuum solvation model with explicit water molecules (CM + water) by Carter et al. [208]. Due to the weak hydrogen bonds between the layers, the basal plane (0001) possesses the lowest surface energy of 0.192 J/m². Such interactions are absent in other surfaces and hence give rise to higher energies of 0.445–0.609 J/m². Thus, the Wulf shape of equilibrium state of β-NiOOH was predicted to have a hexagonal platelet morphology dominated by the (0001) surface. The side walls contain mainly (01$\bar{1}$2) surfaces with a minority of (01$\bar{1}$2) and ($\bar{1}$2$\bar{1}$3) surfaces [208]. In the case of γ-NiOOH, the surface energy of the basal plane (0001) is 0.277 J/m² higher than that of β-NiOOH [210].

Several mechanisms for the OER activity of NiOOH have been studied through DFT calculations [213–217]. These are classified into seven pathways that are summarized below:

A. The "lattice peroxide" path:

$$*H_2O + *O_{latt} \rightarrow *OH + *O_{latt} + H^+(aq) + e^- \tag{47}$$

$$*OH + *O_{latt} \rightarrow *O-O_{latt} + H^+(aq) + e^- \tag{48}$$

$$*O-O_{latt} \rightarrow *\gamma + O_2(g) \tag{49}$$

$$*\gamma + H_2O(l) \rightarrow *O_{latt}H + H^+(aq) + e^- \tag{50}$$

$$*O_{latt}H \rightarrow O_{latt} + H^+(aq) + e^- \tag{51}$$

B. The "hydroperoxide" path:

$$*2H_2O \rightarrow *OH + *H_2O + H^+(aq) + e^- \tag{52}$$

$$*OH + *H_2O \rightarrow *2OH + H^+(aq) + e^- \tag{53}$$

$$*2OH \rightarrow *OOH + H^+(aq) + e^- \tag{54}$$

$$*OOH + H_2O(l) \rightarrow *2H_2O + O_2(g) + H^+(aq) + e^- \tag{55}$$

C. The "electrochemical metal peroxide" path:

$$*H_2O \rightarrow *OH + H^+(aq) + e^- \tag{56}$$

$$*OH \rightarrow *O + H^+(aq) + e^- \tag{57}$$

$$*O + H_2O(l) \rightarrow *OOH + H^+(aq) + e^- \tag{58}$$

$$*OOH + H_2O(l) \rightarrow *H_2O + O_2(g) + H^+(aq) + e^- \tag{59}$$

D. The "oxide" path:

$$2*H_2O \rightarrow 2*OH + 2H^+(aq) + 2e^- \tag{60}$$

$$2*OH \rightarrow *O + *OH + H^+(aq) + e^- \tag{61}$$

$$*O + *OH \rightarrow 2*O + H^+(aq) + e^- \tag{62}$$

$$2*O \rightarrow * + O_2(g) \tag{63}$$

$$* + 2H_2O(l) \rightarrow 2*H_2O \tag{64}$$

E. The "water nucleophilic attack" path:

$$*O_{latt} + H_2O(l) \rightarrow *O_{latt}-OH + H^+(aq) + e^- \tag{65}$$

$$*O_{latt}-OH \rightarrow *O_{latt}-O + H^+(aq) + e^- \tag{66}$$

$$*O_{latt}-O \rightarrow *\gamma + O_2(g) \tag{67}$$

$$*\gamma + H_2O(l) \rightarrow *O_{latt}H + H^+(aq) + e^- \tag{68}$$

$$*O_{latt}H \rightarrow *O_{latt} + H^+(aq) + e^- \tag{69}$$

F. The "lattice oxygen coupling" path:

$$2*O_{latt} \rightarrow *O_{latt}-O_{latt} + *\gamma \tag{70}$$

$$*O_{latt}-O_{latt} + *\gamma \rightarrow 2*\gamma + O_2(g) \tag{71}$$

$$2*\gamma + 2H_2O \rightarrow 2*O_{latt} + 4H+(aq) + 4e^- \tag{72}$$

G. The "bifunctional" path:

$$*H_2O + *O_{latt} \rightarrow *OH + *O_{latt} + H+(aq) + e^- \tag{73}$$

$$*OH + *O_{latt} \rightarrow *O + *O_{latt} + H+(aq) + e^- \tag{74}$$

$$*O + *O_{latt} + H_2O(l) \rightarrow *OO + *O_{latt}H + H+(aq) + e^- \tag{75}$$

$$*OO + *O_{latt}H + H_2O(l) \rightarrow *H_2O + *O_{latt} + O_2(g) \; H+(aq) + e^- \tag{76}$$

Li and Selloni [218] have shown that both β-NiOOH (011$\bar{5}$) and γ-NiOOH (011$\bar{5}$) surfaces are active for OER through the "lattice peroxide" path, where the formation of $*O_{latt}$–O species is RDS. The thermodynamic overpotentials for β-NiOOH and γ-NiOOH were calculated to be 0.46 and 0.52 V, respectively. The most abundant facets of pristine β-NiOOH are the (0001), (101$\bar{2}$), and (1$\bar{2}$1$\bar{3}$) faces, which were studied by Cater et al. using PBE+U and ONIOM-HSE06 methods. The thermodynamic overpotentials, considering various active sites, were calculated to be 0.43–0.48 V [216,219]. The OER mechanism on γ-NiOOH (011$\bar{2}$) was studied by Goddard et al. [220] at the B3PW91 level, considering the implicit solvent. The mechanism follows the "electrochemical metal peroxide" path where the O–O coupling step between the *O adatom and a $H_2O$ molecule is the rate limiting step with an overpotential of 1.21 V. The calculated thermodynamic potential differs by 0.65 eV from the experimental one. This is attributed to: (i) the X-ray crystal structure being different from the model considered in the simulation, (ii) the choice of different calculation methods (seven functionals and four U values) and a different experimental setup, (iii) a variety of mechanistic paths for the sequential proton release.

To find out the precise mechanism, Govind and Carter [216] formulated an automated process where the free energies of various processes are automatically calculated through automatic optimization. Next, it takes the free energies of various processes as inputs and uses the mixed-integer linear programming to find out the OER pathways. The six OER mechanisms on the β-NiOOH (0001) surface with PBE+U functional were performed (Figure 37). The best pathway is the "water nucleophilic attack" path, i.e., * → $*OH_2$ → *OH → *O → *OOH → $*O_2$ → * with an overpotential of η = 0.70 V, which is in agreement with the literature [213].

Defects play a crucial role in developing efficient catalysts for the OER. Based on the synthetic procedure, the NiOOH material can have Ni vacancies, O vacancies, and OH vacancies. The Ni vacancies are present under OER condition and directly favor the process. However, O, and OH vacancies are formed but are quickly repaired by adsorbed water molecules on the surface and can only exist in bulk. Thus, O and OH vacancies indirectly facilitate the OER process. The typical NiOOH motif can maintain its structure intact only in basic condition (pH ~ 14) [221,222]. Thus, it is difficult to use this catalyst under neutral or acidic conditions. Recently, NiOOH was electrochemically synthesized from borate electrolyte ($NiB_i$), which shows good OER activity at pH = 7–9 [206,223,224]. It has two advantages over conventional NiOOH catalysts: self-healing property and smaller particle sizes [145,206,222,224–227]. The thermodynamics of dissolution of Ni was studied and showed that 4.3% Ni undergoes dissolution with an energy change of 0.09 eV. The mechanistic studies show that the thermodynamic overpotential decreases from 0.81 to 0.43 V due to the presence of four-coordinated Ni ($Ni^{4c}$) sites near the Ni

vacancy [211]. It was shown that sub-10 nm $\gamma$-NiOOH particles can provide a good number of Ni vacancies at a neutral condition. Toroker et al. [228] have shown that OH vacancies near the active site of a $\beta$-NiOOH (01$\bar{1}$5) can reduce the overpotential from 0.61 to 0.26 V.

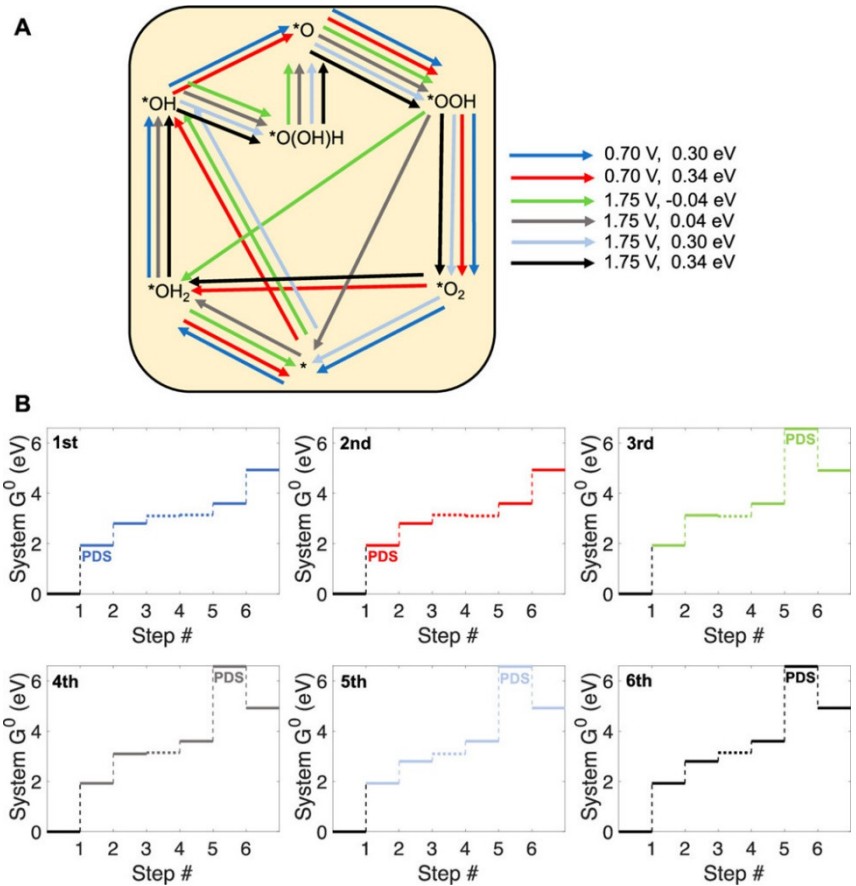

**Figure 37.** (**A**) Schematic of six automatically enumerated OER mechanisms discovered on $\beta$-NiOOH (0001). The corresponding thermodynamic overpotentials (V) and the highest non-electroactive Gibbs free energy steps (eV) are indicated. (**B**) Energetic profiles for six mechanisms by PBE+U ($U_{\text{eff}}$ = 5.5 eV). (Reprinted from ref. [216] with permission from the Journal of Physical Chemistry C. Copyright 2020, American Chemical Society).

Theoretical studies suggest that when NiOOH is doped with Fe, the active site is the exposed Fe instead of Ni, where the coordination number of Fe plays a crucial role. Fe is 4, 5, and 6 coordinated on the bare NiOOH ($\bar{1}$21$\bar{N}$), (011$\bar{N}$), and (0001) surfaces, respectively. Li and Selloni [218] have shown that Fe-doped surfaces $\beta$-NiOOH (01$\bar{1}$5) and $\gamma$-NiOOH (01$\bar{1}$5) follow the "lattice peroxide" path with overpotentials of 0.26 and 0.48 V, respectively. Goddard et al. [220] calculated the thermodynamic overpotential on pure and Fe-doped $\gamma$-NiOOH (01$\bar{1}$2) to be 0.21 and 0.81 V, respectively. The "electrochemical metal peroxide" path was followed on the Fe-doped $\gamma$-NiOOH (01$\bar{1}$2) surface where O–O coupling is the RDS with a kinetic barrier of 0.64 eV. A decrease of the overpotential from 0.67 to 0.45 V was observed by Strasser et al. [229] upon doping Fe on a $\gamma$-NiOOH (01$\bar{1}$0) surface in the "water nucleophilic attack" path. A drop of the overpotential from 0.48 to 0.13 V was reported by Carter et al. [219] on doping $\beta$-NiOOH ($\bar{1}$2$\bar{1}$3) by Fe, which is the lowest reported overpotential in such systems. A "lattice peroxide" path was followed, where the RDS is *Fe–OH $\rightarrow$ *Fe=O + H$^+$(aq) + e$^-$. The robust activity of Fe-doped $\beta$-NiOOH was attributed to stabilization of the terminal oxo species *Fe$^{\text{IV}}$=O and reduction of the adjacent Ni$^{\text{III}}$ by Fe$^{2+}$ [219].

## 4. Discussion and Summary

A variety of computational methods have been developed to understand the OER activity by metal oxides (heterogeneous catalysis), such as coupled cluster methods [230], configurational interactions [231], and quantum chemistry Monte Carlo [232]. In the case of metal oxides, the DFT has been the best choice of studies to understand the structures, semi-conductor properties, the adsorption of reactants, and the successive steps of the reaction mechanism [233]. From DFT, one can find the free energy, enthalpy, and entropy changes in a reaction along with transition state structures.

In metal oxides, to have a concrete prediction of the thermochemistry and kinetics of the reaction, one should have the information of band structures, the electronic structure of oxides, the energies of defects, etc. To understand the redox properties and hence the reaction energy profile, it is essential to have knowledge of the band gap structure of the metal oxide vs. the vacuum. In case of a photoelectrocatalyst evaluation of band gap, the knowledge of the energies of valence and conduction band is important as the from and to the catalyst is dependent on these energies [234].

The properties of metal oxides can be modified by doping other metals. The mixing of metal changes the bonding aspect of M-O bonds and thereby changes either the metal or the oxide, becoming active for the water-splitting reactions. Previous studies have shown a severe problem with DFT while rationalizing catalysis on oxide surfaces due to the huge difference between the molecular orbitals and the excitation energy [235]. In this regard, the Hubbard U correction significantly improves the results [236,237]. Here, all the materials discussed are based on the calculation of U-type corrections.

The formation of hydroxide bound surface (OH*) is the key step in $TiO_2$ [49]. The overpotential is the lowest, with a total O covered $S^2$ surface, although higher than $RuO_2$ and $IrO_2$. This is due to the different occupations of the nearest-neighbor CUSs [55]. The rate-limiting step is determined by the surface-reaching photo-holes ($C_{h+}$). The different surface structures of anatase (101), (001), and (102) concluded that local surface structures of anatase have little effect on the OER mechanism [89]. The reactivity of different structures follows the order: of anatase $\approx$ brookite > rutile [58].

The commonly known polymorphs of $MnO_2$; $\alpha$- (tetragonal), $\beta$- (tetragonal), and $\delta$-(triclinic) are poorly active to OER activity. However, recently developed electrochemically synthesized induced layer $\delta$-$MnO_2$ (EI-$\delta$-$MnO_2$) shows comparable performance to precious metal oxides (Ir/$RuO_x$). EI-$\delta$-$MnO_2$ possesses a special edge site with neighboring Mn vacancy [60]. For the pristine spinel $Mn_3O_4$ (001) and $\delta$-$MnO_2$ (2116), the lowest energy pathway follows the $H_2O \rightarrow OH \rightarrow O \rightarrow O_2$ mechanism. The electrocatalytic activity is enhanced by stabilizing metastable or non-native (NN) polymorphs of $MnO_2$ [130]. Among $\beta$/N-, $\gamma$/N-NN1-, r/NN1-, $\alpha$/NN2-, and $\delta$/NN3-$MnO_2$ polymorphs, the higher specific activity of $\delta$/NN3-$MnO_2$ is attributed to the low oxidation state (+3.5) of Mn. The $\alpha$ phase $MnO_2$ exhibits superior OER catalytic performance compared to $\delta$-$MnO_2$. The metal oxides with smaller band gap show higher electrical conductivity and better OER properties through the transfer of electrons [132,133]. The hollandite crystal structure of $\alpha$-$MnO_2$ shows better catalytic activity compared to the other polymorphs.

In the cobalt oxide catalyst, cubane-like Co(IV)-oxo unit species are formed at the surface of the catalyst [65]. The calculated Pourbaix diagram showed that the CoOOH phase is the most stable phase under typical OER conditions (pH = 12–13 and $U$ > 1.23 to ∼1.7 V vs. RHE) [68]. Among (0001), (01$\bar{1}$2), and (10$\bar{1}$4) surfaces of $\beta$-CoOOH, the (10$\bar{1}$4) is more active as Co is in +3 oxidation state while it is +4 in other surfaces. The overpotential for OER is further decreased by doping with Ni to form $Ni_yCo_{1-y}O_x$ thin films. From the studies of (110) surface of the spinel cobalt oxide, $Co_3O_4$, it was shown that the deprotonation of adsorbed OH* to form O* is the more challenging step [161]. The advantages of $Co_3O_4$ for catalytic applications are its availability, stability [168], and p-type semiconducting property (low conductivity) [169].

Huang et al. [197] demonstrated that due to structural deformities, the electron configuration of $Co^{III}$ in ultrathin CoOOH nanosheets changes from $t^6_{2g}e^0_g$ to $t^5_{2g}e^{1.2}_g$, ultimately

giving enhanced OER activity. Li et al. [198] have shown through DFT calculations that strain engineering can be utilized to enhance the OER activity by two-dimensional (2D) CoOOH. It was shown (Figure 32) that by increasing the tension by 9%, the electronic configuration of Co$^{III}$ changes from low spin (LS: $t^6_{2g}e^0_g$) to high spin (HS: $t^4_{2g}e^2_g$). LS CoOOH is a poor catalyst as it requires large energy (1.35 eV) for the release of $O_2$.

A comparative stability study was performed by Chen et al. [199] for crystalline cobalt oxides and hydroxides: $CoO$, $Co(OH)_2$, $Co_3O_4$, $CoO(OH)$, and $CoO_2$ (Figure 34) in an electrochemical environment using DFT with on-site Coulomb repulsion. Under low pH reducing conditions ($V < 0$ vs. SHE), $Co(OH)_2$ is thermodynamically stable, while at high pH and potential ($V < 1.23$ eV vs. SHE), $CoO(OH)$ and $CoO_2$ are stable. They showed corroborating experimental results on the (0001) natural cleavage surface of $CoO(OH)$ in the presence of $CoO_2^{x-}$ ($x$ = 0–0.5) layer under oxidizing conditions. In the OER mechanism, the layered oxide transformation of adsorbed water to OH* is the energetically most demanding process and leads to high overpotential. Thus, layered $CoO_2^{x-}$ is not active for the OER compared to active $Co_3O_4$ where the O* $\rightarrow$ OOH* is the rate-determining step with a smaller barrier. Analysis of the electronic properties shows that an additional Fe layer stabilizes the balk-like oxidation state of +2 for Co and Ni at the A-layer.

The OER activity was comparatively investigated in β-NiOOH(01$\bar{1}$5), γ-NiOOH(101), Fe-doped β-NiOOH(01$\bar{1}$5), and γ-NiOOH(101), $NiFe_2O_4$(001), and the $Fe_3O_4$(001) surface as electrocatalysts for OER [206]. It was found that Fe-doped β-NiOOH(01$\bar{1}$5) is the most active, with an overpotential (η = 0.26 V) lower than that of $RuO_2$ (0.36 V) [50,63], followed by $NiFe_2O_4$(001) (η = 0.42 V). The trend of the overpotential follows the order: Fe-doped β-NiOOH (0.26 V) < $NiFe_2O_4$ (0.42 V) < β-NiOOH (0.46 V) < Fe-doped γ-NiOOH (0.48 V) < γ-NiOOH (0.52 V) < $Fe_3O_4$ (0.70 V). In Fe-doped γ-NiOOH(101), two adjacent water molecules lose two protons subsequently.

NiOOH behaves as a semiconductor with a band gap of 1.7–1.8 eV. It was stated from the PBE+U calculations that no band gap in the spin-up component (majority) and hence a half-metallic ferromagnetic (FM) configuration, is prevalent [207]. A band gap of 1.5 and 2.5 eV is calculated at the hybrid HSE06 and PBE0 functionals. By implementing $G_0W_0$ calculations, Carter et al. reported a band gap of 1.96 eV staggered-II β-NiOOH, corroborating well with the PBE/HSE06 results. In an octahedral coordination environment, the electronic configuration $Ni^{(2+n)+}$ is $t_{2g}^6e_g^{2-n}$ was suggested from the crystal field theory. Through a benchmark study using PBE, PBE+U, DFT-vdW (optPBE, optB86, and optB88), hybrid functionals (HSE06 and PBE0), and DFT-D2 (HSE06-D2 and PBE0-D2) functionals, it was shown that PBE+U functional is sufficient [212].

Different plausible pathways of OER—lattice peroxide, hydroperoxide, electrochemical metal peroxide, oxide, water nucleophilic attack, lattice oxygen coupling, and bifunctional—are discussed using the NiOOH material. It was shown that β-NiOOH (01$\bar{1}$5) and γ-NiOOH (01$\bar{1}$5) surfaces are active for OER through the "lattice peroxide" path, where the formation of *Olatt–O species is RDS [198].

Govind and Carter developed an automated process [196] to find the precise mechanism where the reaction free energies are automatically calculated using mixed-integer linear programming to find the OER pathways. Among the six plausible pathways studied on the β-NiOOH (0001) surface, the "water nucleophilic attack" path was found to be the best, which agrees with the experimental report. Ni vacancies favoring the OER mechanism are present. The active site is the exposed Fe instead of Ni, when NiOOH is doped with Fe. On the bare NiOOH ($\bar{1}$2$\bar{1}\bar{N}$), (011$\bar{N}$), and (0001) surfaces, Fe is 4, 5, and 6 coordinated, respectively. It was shown that in β-NiOOH (01$\bar{1}$5) and γ-NiOOH (01$\bar{1}$5), a "lattice peroxide" path is favored. On the Fe-doped γ-NiOOH (01$\bar{1}$2) surface, the "water nucleophilic attack" path is favored.

The energetics of the intermediates are typically taken into account while examining the heterogenous OER mechanisms using computational approaches. Now, the question is how the situation would be affected if the transition state is considered. An ideal catalyst would proceed through the required four steps with a free energy increase of 1.23 eV at

$U = 0$ V. The catalyst surface would be covered uniformly with all types of possible intermediates. Under this condition, a single-charge-transfer reaction of OER is considered as the RDS concept is invalid. A transition state is ignored in simple DFT calculation. However, considering the transition state, the energy barrier would be higher [238]. Thus, the actual free reaction barrier is underestimated in simple DFT calculation with intermediate energy.

## 5. Conclusions

In recent years scientists have questioned the homogeneous water oxidation process, as most metal complexes under high potential and in alkaline media decompose to oxide materials. Thus, it is really a question of whether the complex or the metal oxide is the active catalyst. Furthermore, the durability and applicability of heterogeneous catalysts is more than that of the homogeneous ones. Thus, it is of high importance to develop cheap metal oxides for industrial applications of the water splitting reaction. In this study we have reviewed widely studied earth abundant first row transition metal oxides/hydroxides: titanium oxide ($TiO_2$), manganese oxides/hydroxides ($MnO_x$/MnOOH), cobalt oxides/hydroxides ($CoO_x$/CoOOH), and nickel oxides/hydroxides ($NiO_x$/NiOOH).

$TiO_2$ is an important photo electrocatalyst for the water oxidation reaction (OER). The density functional theory (DFT) based calculations on rutile $TiO_2$ (110), anatase $TiO_2$ (101), and brookite $TiO_2$ (210) model surfaces were discussed. The selectivity towards the formation of $^{\bullet}OH$, $H_2O_2$, or $O_2$ are studied through the construction of potential-dependent free energy diagrams of the water oxidation process.

Manganese-based catalysts are always interesting to study due to their existence in photosystem II. It was shown that electrochemically synthesized $\delta$-$MnO_2$ (EI-$\delta$-$MnO_2$) materials show comparable performance to precious metal oxides. However, the stable polymorphs: $\alpha$- (tetragonal), $\beta$- (tetragonal), and $\delta$-(triclinic), are poor in their OER activity. It was demonstrated that the EI-$\delta$-$MnO_2$ with a special edge site with a neighboring Mn vacancy provides excellent OER activity using first-principles calculations with an overpotential of 0.59 V, 0.19 V, lower than that of pristine $MnO_2$. The phase diagram of the transformation of $Mn_3O_4$ to $H_xMnO_2$ and to $MnO_2$ at pH = 7 was discussed. The three different local structures: adatom, dislocation, and edge-NV, as obtained in the solid phase transition mechanism, were examined during OER activity.

A lot of theoretical work has been carried out to understand the structure and catalytic activity of NiOOH. Ni present in a mixed oxidation state of $Ni^{3+}$ and $Ni^{4+}$ in NiOOH and their ratio varies from phase to phase. Hence, in understanding the mechanism, all possibilities in terms of oxidation should be taken into consideration. The presence of new $NiO_2$ frameworks in tunneled NiOOH was suggested. To screen the bulk crystal structure of $\beta$-NiOOH (with the variation of H from 0 to 2), machine learning global PES optimization was utilized. An H concentration below 0.5 O-O peroxide bond formation was observed in a tunneled $NiO_2$ framework. Further O concentration may also change during the catalysis. In the "water nucleophilic attack" path, the intermediate O atom can be inserted within the lattice to form "lattice peroxide", and hence change the structure. The accurate energies and band structures cannot be obtained with DFT+U, DFT-vdW, and even hybrid functionals. To verify the result, the post-DFT hybrid HSE06 should be employed.

Finally, it should be pointed out that even though the general mechanisms of the OER processes on all the oxides discussed are similar, the detailed mechanisms differ considerably. The detailed evaluation of each mechanism is complicated by the observation that the OER process affects the surface structure of the catalyst.

**Author Contributions:** Conceptualization, S.G.P. and D.M.; methodology, S.G.P.; validation, S.G.P. and D.M.; formal analysis, S.G.P.; investigation, S.G.P.; resources, S.G.P. and D.M.; data curation, S.G.P.; writing—original draft preparation, S.G.P.; writing—review and editing, S.G.P. and D.M.; visualization, S.G.P.; supervision, S.G.P. and D.M.; project administration S.G.P. and D.M. All authors have read and agreed to the published version of the manuscript.

**Funding:** This research received no external funding.

**Informed Consent Statement:** Not applicable.

**Acknowledgments:** SGP thanks IIT Kharagpur for post-doctoral fellowships. SGP thanks Professor Guangchao Liang for kindly inviting to contribute to the special issue "Computational Catalysis: Methods and Applications."

**Conflicts of Interest:** The authors declare no conflict of interest.

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
