# Peer review of "On the Mechanism of Heterogeneous Water Oxidation Catalysis: A Theoretical Perspective"

_inorganics, doi:10.3390/inorganics10110182_

Round 1

Reviewer 1 Report

The manuscript titled "On the Mechanism of Heterogeneous Water Oxidation Catalysis: A Theoretical Perspective" by Patra et al., reported a comprehensive summary of the water oxidation mechanism.  Mostly, they focus on the abundant transition metal oxides, which are potentially cost-effective for large-scale applications. The crystalline structure and surface intermediates are systematically compared, such as TiO2, MnOx, CoOx, NiOx, CoOOH, NiOOH etc. Moreover, they compared the energy barriers between each step/intermediate/transient state, and the rate-limiting steps were investigated. Overall, this is a nice work, I would like to suggest a major revision a this stage. The questions are listed as follows:

1. In the first part, there were serious errors in the potentials.  From most of the handbooks or literatures, the standard potential for O2/H2O was not -1.23 V vs. NHE in acidic solution, nor -0.40 V vs. NHE in basic solution. Moreover, the theoretical voltage of two-electrode cells was not -1.23 V vs. NHE.  The authors should understand the potential was defined relative to the reference.  

2.  Fig. 1 and 2 are two typical schemes for water oxidation, which is quite different from other examples in the latter.  It is necessary to cite references, the suitable conditions.

3. Compared with the generous electrocatalysts, TiO2, Fe2O3 or WO3 are also semiconductors. The authors should specify the advantages/disadvantages, when they are applied as water oxidation electrocatalysts.

4. The authors have listed many crystalline structures for water oxidation. It is better to highlight which could be the potential or excellent one for water oxidation.

5. Moreover, the band structures were summarized, which were usually calculated by DFT methods.  It is better to highlight the relationship between this band structure and the activity for water oxidation.

6. It is well known that transition metal oxides, such as Mn, Co, Ni OOH compounds are good electrocatalyst, which is already summarized in many reviews. The authors should specify the new findings in this manuscript.

7. The discussion and perspective are poorly prepared. Authors should give the lessons or their suggestions to the readers.

Author Response

Reviewer 1

The manuscript titled "On the Mechanism of Heterogeneous Water Oxidation Catalysis: A Theoretical Perspective" by Patra et al., reported a comprehensive summary of the water oxidation mechanism.  Mostly, they focus on the abundant transition metal oxides, which are potentially cost-effective for large-scale applications. The crystalline structure and surface intermediates are systematically compared, such as TiO2, MnOx, CoOx, NiOx, CoOOH, NiOOH etc. Moreover, they compared the energy barriers between each step/intermediate/transient state, and the rate-limiting steps were investigated. Overall, this is a nice work, I would like to suggest a major revision a this stage. The questions are listed as follows:

  1. In the first part, there were serious errors in the potentials.  From most of the handbooks or literatures, the standard potential for O2/H2O was not -1.23 V vs. NHE in acidic solution, nor -0.40 V vs. NHE in basic solution. Moreover, the theoretical voltage of two-electrode cells was not -1.23 V vs. NHE.  The authors should understand the potential was defined relative to the reference.  

These are corrected. Please check page 2.

  1. Fig. 1 and 2 are two typical schemes for water oxidation, which is quite different from other examples in the latter.  It is necessary to cite references, the suitable conditions.

Relevent references are cited. Please see caption of Figures 1, and 2, highlighted in yellow (references 71, and 72 in the revised MS).

  1. Compared with the generous electrocatalysts, TiO2, Fe2O3 or WO3 are also semiconductors. The authors should specify the advantages/disadvantages, when they are applied as water oxidation electrocatalysts.

A comparison has been added to the revised manuscript. Please see page 15 highlighted in yellow.

  1. The authors have listed many crystalline structures for water oxidation. It is better to highlight which could be the potential or excellent one for water oxidation.

When we discuss the specific material it is clearly mentioned in the result & disvussion part. In the revised manuscript a summary section has been added. There also we have highlighted the most active crystalline form.

  1. Moreover, the band structures were summarized, which were usually calculated by DFT methods.  It is better to highlight the relationship between this band structure and the activity for water oxidation.

These are discussed in the relevant sections. Section 4, pages 42-44 in revised MS, highlighted in yellow.

  1. It is well known that transition metal oxides, such as Mn, Co, NiOOH compounds are good electrocatalyst, which is already summarized in many reviews. The authors should specify the new findings in this manuscript.

I agree that researchers have discussed these materials but a complete discussion based on these earth abundant material (with emphasis on Ti, Mn, Co, and Ni) focusing the theoretical aspect is not present in the literature. Now we have added a summary section where we have emphasized the relevance of the article (Section 4, pages 42-44 in revised MS, highlighted in yellow).

  1. The discussion and perspective are poorly prepared. Authors should give the lessons or their suggestions to the readers.

To the revised manuscript methodology section has been added. More discussion have been added in different sections of the revised MS.

Reviewer 2 Report

In this work, authors summarized the current understandings of the reaction mechanism associated with oxygen evolution reaction catalyzed by Ti, Mn, Co, and Ni oxide/hydroxide. The focus seems to put on the outcome from computational simulation. Although there are substantial materials in the manuscript, it is hard to get a clear and consistent picture about each kind catalyst introduced by authors. Somehow, it feels like a lot of stuff are stacking together without a meaningful connection. It is recommended to publish after a major revision. 

1.     The abstract is difficult to understand and is not very appropriate for a review article. 

2.     The main discussions are related with first principle calculations as title suggested. It might be helpful to have a section to introduce basic methodology of state of art of computational method which could help a broad readership.

3.     For each introduced materials, it might be useful gave a general background description at the beginning and brought up the current remaining issues which can be solved or assisted by computational calculations. Then it organized corresponding studies to show these aspects which will gave a good summary. 

Author Response

Reviewer 2

In this work, authors summarized the current understandings of the reaction mechanism associated with oxygen evolution reaction catalyzed by Ti, Mn, Co, and Ni oxide/hydroxide. The focus seems to put on the outcome from computational simulation. Although there are substantial materials in the manuscript, it is hard to get a clear and consistent picture about each kind catalyst introduced by authors. Somehow, it feels like a lot of stuff are stacking together without a meaningful connection. It is recommended to publish after a major revision. 

  1. The abstract is difficult to understand and is not very appropriate for a review article. 

The abstract has been modified.

  1. The main discussions are related with first principle calculations as title suggested. It might be helpful to have a section to introduce basic methodology of state of art of computational method which could help a broad readership.

The methodology section has been added to the revised manuscript. Please see Section 2, pages 3-4 of the revised MS, highlighted in yellow.

  1. For each introduced materials, it might be useful gave a general background description at the beginning and brought up the current remaining issues which can be solved or assisted by computational calculations. Then it organized corresponding studies to show these aspects which will gave a good summary. 

A background discussion has been added to the discussion of Co and Mn oxide material. For Ti, and Ni some discussions are already present.

Reviewer 3 Report

In this work the authors presented comprehensive review article on the theoretical modelling of OER on state-of-the-art catalysts.

The work is well written and summarizes recent achievements in the field of electrocatalysis of OER. Therefore I am positive for publication of this work, and I have only a few aspects that the authors should address.

1) The choice of the systems object of the study should be highlighted better, as well as the link between one application to the other.

2) The description is very detailed. The readership could find useful some take-home messages for each material studied, as well as challenges in the field.

3) I suggest the authors to introduce the problem of modelling OER on a catalytic material before going to the applications. This implies to expand Section 2, highlighting the various possible path that can be followed by the reaction. This will be of help also at page 34-35.

4) Related to 2, the problem of the “reaction path” and formation of unconventional intermediates is a certain challenge for theoretical predictions. This is particularly important for any catalyst, from oxides as RuO2 [Energy & Environmental Science 10.12 (2017): 2626-2637; Nature Catalysis 3.6 (2020): 516-525] to Single-Atom Catalysts (SACs) [Journal of the American Chemical Society 143.48 (2021): 20431-20441; ACS Catalysis 12 (2022): 11682-11691; ACS Catalysis 10.7 (2020): 4313-4318]. Altought SACs are not a topic of this work, they are probably the systems more sensitive to this aspect. I suggest the authors to discuss this challenge in the field.

Author Response

Reviewer 3

In this work the authors presented comprehensive review article on the theoretical modelling of OER on state-of-the-art catalysts.

The work is well written and summarizes recent achievements in the field of electrocatalysis of OER. Therefore, I am positive for publication of this work, and I have only a few aspects that the authors should address.

1) The choice of the systems object of the study should be highlighted better, as well as the link between one application to the other.

In the revised manuscript the discussion and background have been emphasized in details.

2) The description is very detailed. The readership could find useful some take-home messages for each material studied, as well as challenges in the field.

For this a summary section has been added.

3) I suggest the authors to introduce the problem of modelling OER on a catalytic material before going to the applications. This implies to expand Section 2, highlighting the various possible path that can be followed by the reaction. This will be of help also at page 34-35.

This has been added to the revised MS, please see pages 7-8, highlighted in yellow.

4) Related to 2, the problem of the “reaction path” and formation of unconventional intermediates is a certain challenge for theoretical predictions. This is particularly important for any catalyst, from oxides as RuO2 [Energy & Environmental Science 10.12 (2017): 2626-2637; Nature Catalysis 3.6 (2020): 516-525] to Single-Atom Catalysts (SACs) [Journal of the American Chemical Society 143.48 (2021): 20431-20441; ACS Catalysis 12 (2022): 11682-11691; ACS Catalysis 10.7 (2020): 4313-4318]. Altought SACs are not a topic of this work, they are probably the systems more sensitive to this aspect. I suggest the authors to discuss this challenge in the field.

These are discussed in pages 7-8 and relevant references are cited in the context.

Round 2

Reviewer 1 Report

The authors have tried to respond to the comments and revised their manuscript appropriately.  There are two minor comments:

(1) there are review papers, which is relevant to this work. For instance, the reviewer published on Nano Research (2021, 14, 3446).

(2) The authors have compared the various transition metal oxides, using the DFT methods (e.g., by J. Norskov). The pros and cons could be compared with the transient state, as the free energy change of the intermediates may not be the same as the active energies.

Author Response

Reviewer 1

The authors have tried to respond to the comments and revised their manuscript appropriately.  There are two minor comments:

(1) there are review papers, which is relevant to this work. For instance, the reviewer published on Nano Research (2021, 14, 3446).

The relevant reference has been cited. Please see reference 244, highlighted in yellow.

(2) The authors have compared the various transition metal oxides, using the DFT methods (e.g., by J. Norskov). The pros and cons could be compared with the transient state, as the free energy change of the intermediates may not be the same as the active energies.

A discussion has been added. Please see page 44, highlighted in yellow.

Reviewer 2

It can be accepted in current form.

We thank the reviewer for accepting the manuscript.

Reviewer 2 Report

It can be accepted in current form. 

Author Response

We thank the reviewer for accepting the manuscript.
